# Short-term cooling, drying and deceleration of an ice-rich rock glacier

Alexander Bast[1,2], Robert Kenner[1,2], Marcia Phillips[1,2]

[1]Alpine Environment and Natural Hazards / Permafrost, WSL Institute for Snow and Avalanche Research SLF, 7260 Davos Dorf, Switzerland

[2]Climate Change, Extremes and Natural Hazards in Alpine Regions Research Center CERC, 7260 Davos Dorf, Switzerland

*Correspondence to*: Alexander Bast (alexander.bast@slf.ch)

**Abstract.** Observations in the European Alps show a long-term rise in rock glacier velocities, which is often associated with increased air and ground temperatures and, more recently, water content. Long-term rock glacier acceleration is superimposed

by a high interannual variability of the velocity with a particular gap in the quantitative assessment of the role of water in rock glacier kinematics and the factors leading to short-term rock glacier deceleration.

To address this research gap, we drilled three vertical boreholes in the Schafberg rock glacier, Swiss Alps, in August 2020. We documented their stratigraphy and equipped one of the boreholes with temperature sensors and piezometers, and the other two with cross-borehole electrodes for electrical resistivity tomography measurements. Rock glacier velocities were

determined using repeated terrestrial laser scans. Using data from an additional borehole and nearby weather stations and ground surface temperature sensors we analyzed the interactions between meteorological and subsurface conditions during a rock glacier deceleration period, from January 2021 to July 2023.

Our findings show that a lowering of the water content in rock glacier shear horizons is crucial for interannual rock glacier deceleration. The impact of the snowpack, both as an insulator and as a water source is significant for rock glacier kinematics.

Winters with little snow and relatively dry summers appear to be ideal for rock glacier cooling and drying, leading to deceleration. Summer heat waves have limited impact on rock glacier velocity if they are preceded by snow-poor winters.

Our study uses an innovative combination of borehole data to gain insights into rock glacier temperatures and water contents, allowing to detect relative changes in ice and/or water contents in ice-rich permafrost. The monitoring techniques used have the potential to contribute to a better understanding of the main drivers of rock glacier kinematics and water availability.

**1 Introduction**

A multi-decadal and significant increase in rock glacier displacement velocities has been observed in the Swiss Alps (Micheletti et al., 2015; Roer et al., 2005; Permos, 2023b) and other European mountain ranges (e.g., Fleischer et al., 2021; Thibert and Bodin, 2022; Eriksen et al., 2018; Kellerer-Pirklbauer et al., 2024). The movement of rock glaciers basically involves two components: (i) creep, indicating plastic deformation in the ice-rich substrate, and (ii) shearing within the shear

horizon typically found in ~ 12 m to ~ 32 m depth (Arenson et al., 2002). The shearing component is expected to be the primary contributor to the overall deformation (Arenson et al., 2002), and hence to rock glacier kinematics.

Rock glacier kinematics are driven by common external climatic factors, and they vary over time (Delaloye et al., 2010b; Delaloye et al., 2008), with phases of acceleration, interrupted by periods of stagnation or deceleration of variable duration (Permos, 2023b). Rock glacier deceleration occurs on a seasonal basis (Wirz et al., 2016), with variable timing on different rock glaciers (Delaloye et al., 2010a). However, rock glacier deceleration can also last over several seasons/years (Permos, 2023b). Interannual rock glacier deceleration notably occurs after snow-poor autumns and winters, during which the ice-rich permafrost can cool efficiently, and groundwater can freeze (Kenner et al., 2020). Whilst numerous studies focus on quantifying rock glacier acceleration (Micheletti et al., 2015), investigating the time scales involved (Wirz et al., 2016; Bertone et al., 2023) and identifying its drivers (Cicoira et al., 2019a), only few studies mention the conditions leading to rock glacier deceleration (e.g., Ikeda et al., 2008; Thibert and Bodin, 2022; Bearzot et al., 2022; Kellerer-Pirklbauer and Kaufmann, 2012).

Analyses of rock glacier kinematics are often carried out using remote sensing data, which, for instance in the case of spaceborne interferometric synthetic aperture radar (InSAR), even allow quantification of the rate and direction of rock glacier creep at a global scale (Bertone et al., 2022). On a local scale, drone-based and terrestrial remote sensing techniques, such as LiDAR (Light Detection and Ranging), can cover entire landforms (Vivero et al., 2022). In-situ displacement monitoring sensors such as GNSS (global navigation satellite system) deployed over selected parts of rock glaciers provide ground truth with a high temporal resolution (Cicoira et al., 2022). Linking remote sensing data, water input, and subsurface data is essential to understand the drivers of rock glacier kinematics and to unravel the complex links between external drivers and the internal characteristics of a rock glacier. Recent literature highlights that air and ground temperature as well as water content are key parameters determining rock glacier kinematics (Cicoira et al., 2019b; Cicoira et al., 2019a). The timing of the seasonal snow cover, its depth and duration on and in the surroundings of rock glaciers are of particular importance, as snow is a temperature regulator and a source of meltwater (Kenner et al., 2020).

When moving from rock glacier kinematics to their internal characteristics (Arenson et al. 2002), direct subsurface data from rock glaciers is quite scarce, for logistic and financial reasons. Borehole temperatures are currently the most widespread type of subsurface data, allowing to monitor the temperature regime, active layer thickness/duration and thermal anomalies, such as those triggered by water or air fluxes (Noetzli et al., 2021; Permos, 2023b; Luethi and Phillips, 2016; Zenklusen Mutter and Phillips, 2012b). Nevertheless, temperature data alone do not allow to discern between ice and water, which can coexist at 0 °C and information on water content is essential to understand rock glacier kinematics. Applied near-surface geophysics such as electrical resistivity tomography (ERT), refraction seismic tomography (SRT), ground penetrating radar (GPR) and electromagnetic methods deliver valuable information on rock glacier internal structure and the distribution of rock, air, ice and water (Hauck, 2013; Boaga et al., 2020; Pavoni et al., 2023; Hauck et al., 2011; Kneisel et al., 2008). Cross-borehole geophysics provide higher-resolution information on the near-surface structure (Binley and Slater, 2020) and are a means to distinguish ice and water in ice-rich permafrost and to monitor their volumetric variation and distribution over time by, e.g., electromagnetic velocity structures (GPR; traveltime tomography; Musil et al., 2006) or detecting changes with inverted resistivity models (Musil et al., 2006; Phillips et al., 2023).

This paper aims to understand variations in rock glacier kinematics and the conditions leading to rock glacier deceleration. We focus on the slowing of the Schafberg Ursina III rock glacier (Pontresina, Eastern Swiss Alps) in the 2.5 years from January 2021 to July 2023. A combination of borehole temperature, piezometric pressure, cross-borehole ERT (Phillips et al., 2023), and terrestrial laser scan data (Kenner et al., 2020) together with meteorological data are used to analyze the interaction between meteorological and subsurface conditions during a multiseasonal rock glacier deceleration. The period analyzed was affected by low precipitation values and a summer heat wave, during which the air temperatures recorded between June and September 2022 were significantly higher than in June–September of the past decade (Cremona et al., 2023).

## 2 Material and Methods

### 2.1 Study site and field setup

The Schafberg Ursina rock glacier complex (46°29′50.391″ N, 9°55′34.779″ E) is an ice-rich, creeping landform located in the Eastern Swiss Alps between ∼ 2700 and ∼ 2900 m asl above Pontresina. It consists of three distinct rock glaciers, Ursina I, II and III in a west-facing cirque (Fig. 1). The rock glacier complex is surrounded by three peaks and steep rock faces consisting mainly of banded biotite gneiss of the Austroalpine nappes (Peters, 2005). The regional climate has a central alpine character with relatively low precipitation and a comparatively high air temperature amplitude (Ott, 1997).

Destructive drillings in 1990 (B1) (Vonder Mühll and Holub, 1992) and 2020 (B3 to B5) (Phillips et al., 2023) at Ursina III rock glacier (Fig. 1e) show that the uppermost 3 – 4 m consist of large boulders (Fig. 2), below which icy sediments and dirty ice are found (B3, Fig. 2). In two boreholes drilled in 2020 (B4 & B5) wet sludge with ice dominates below the blocky layer. Ice and water distribution are heterogeneous over distances of 5 to 10 m (Phillips et al., 2023). In 1990, bedrock was reached at a depth of 16 m (Vonder Mühll and Holub, 1992). In 2020, bedrock was not reached (Fig.2).

Various sensors are deployed on and in the rock glacier (Fig. 1), allowing to monitor ground temperature (borehole thermistors), pore water pressure (borehole piezometers), ground resistivities (cross-borehole ERT), ground surface temperature (miniature temperature data loggers), and meteorological conditions (weather station). Ground surface displacements are measured annually using a terrestrial laser scanner (TLS). We focus our analysis on borehole data collected between 01 January 2021 and 30 June 2023. We include meteorological, GST and TLS data from 2020 onwards to improve the interpretation.

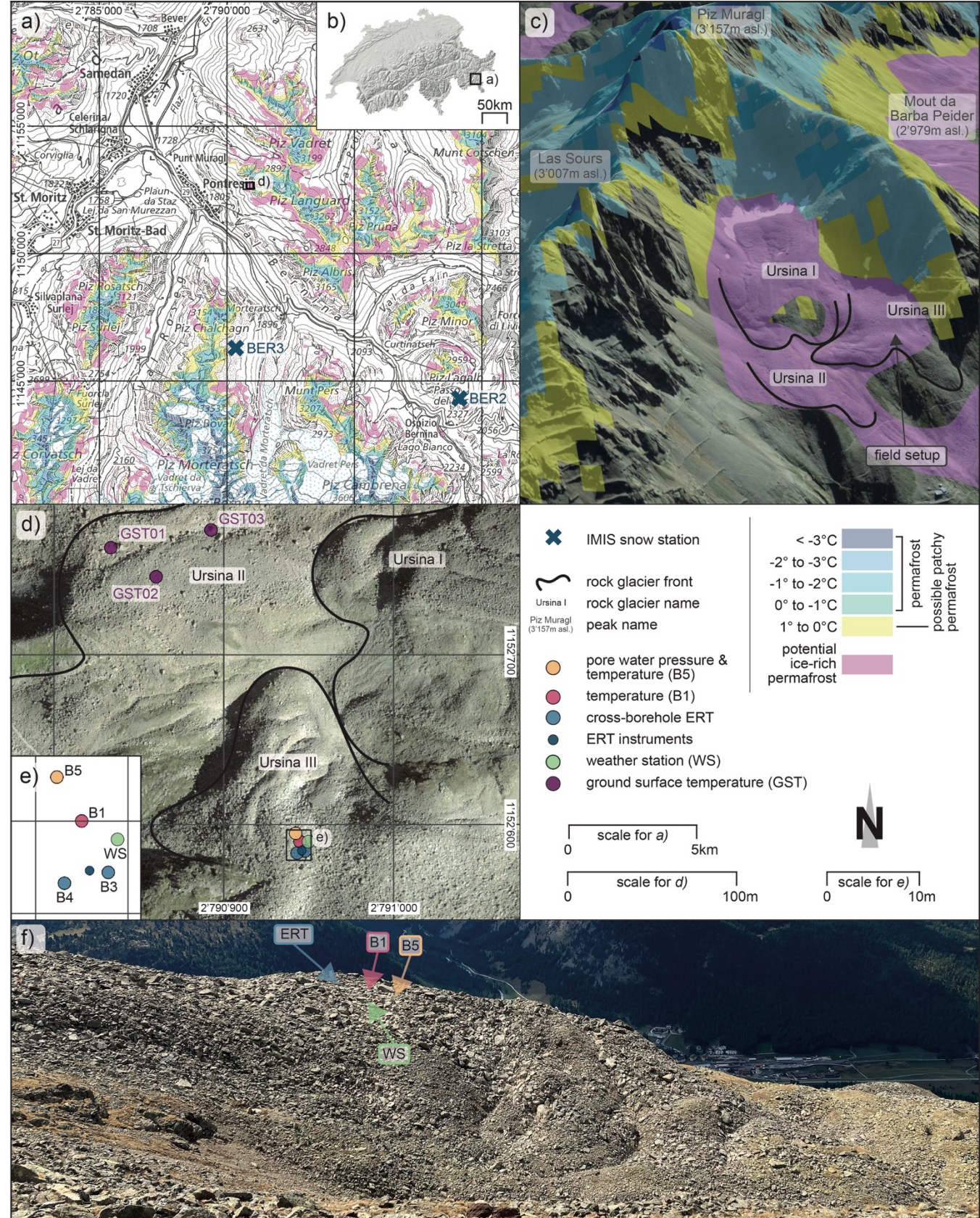

**Figure 1: The Schafberg study site (d) is located in the Upper Engadine (a), Canton of Grisons, Switzerland (b). Blue bold crosses mark the IMIS weather stations Valetta (BER2) and Puoz Bass (BER3) (a). The Schafberg rock glacier complex comprises the lobes Ursina I, II, and III, and is visualised in the 3D scene (c) together with the surrounding rock faces, the three highest peaks (Piz Muragl, Las Sours, Mout da Barba Peider), and the modelled permafrost distribution according to Kenner et al. 2019 (available at maps.wsl.ch). The orthophotograph and inset map provide an overview of the field setup with boreholes and corresponding instruments/sensors. The image (f) shows the Ursina III lobe, featuring a distinct rock glacier front and a well-defined ridge and furrow system. The surface consists of large blocks and boulders. The arrows point the location of the field-setup, also visible in (d) and (e). The north arrow is valid for all subplots. Base maps (a, b, and d) are provided by swisstopo, while the 3D scene with the permafrost map can be accessed at maps.wsl.ch.**

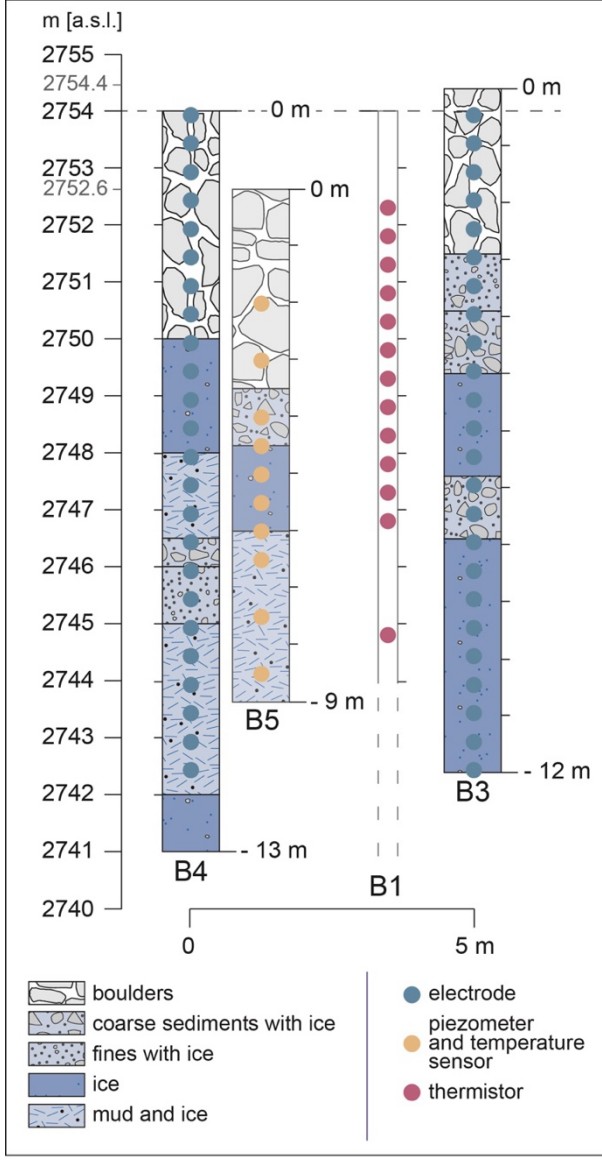

**Figure 2: Stratigraphy of the boreholes B3, B4 and B5 as recorded during the drilling in August 2020. The stratigraphy of B1, drilled in 1990, is not available. Filled dots indicate the depths of various sensors within the boreholes, including the electrode positions of**

the cross-borehole electrical resistivity tomography (ERT) (ERT multicore cable with stainless steel electrodes; marked in blue), piezometers and thermistors (Keller PAA-36XIW with PT 1000 temperature sensor; marked in yellow), and thermistors (YSI thermistors 44006; marked in pink). The figure has been adapted from Phillips et al. 2023.

## 2.2 Meteorological variables and ground surface temperature data

Our study investigates the effect of meteorological forcing (air temperature, snow and rainfall) on ground temperature, ground water pressure, ground electrical resistivity, and horizontal rock glacier displacement rates. To describe the meteorological conditions, we used data from the two nearby automated weather stations Puoz Bass (BER3) and Valetta (BER2) of the Intercantonal Measurement and Information System (IMIS network) (Imis, 2023), and an all-in-one weather station at Schafberg (WS), which we installed on 04 August 2022. Table S1 provides a detailed overview of the locations, sensors and corresponding meteorological variables with computed descriptive statistics.

As the snow and weather stations use unheated rain gauges, we calculated liquid precipitation (rainfall, RA) using a 4 °C air temperature threshold, above which the probability of snowfall is very low (Jennings et al., 2018; Rohrer, 1989). We use modelled snow water equivalent (SWE) to quantify the liquid water stored within the snowpack. SWE values were provided by the IMIS network, where the physically based model SNOWPACK is used to calculate snow cover evolution (Bartelt and Lehning, 2002; Lehning et al., 2002a; Lehning et al., 2002b). There are no direct snow recordings on the Schafberg rock glacier. We used near-surface ground temperature data from three temperature sensors (GST) distributed across the Ursina II lobe (Fig. 1) and buried about 10 cm below the ground surface (Permos, 2023b). These delivered indirect information on the duration of the snow cover using descriptive temperature variations and zero curtain analysis (Staub & Delaloye 2017).

For data visualization, we aggregated air temperature (TA; daily, weekly, and monthly arithmetic averages), RA (daily and monthly sums), snow height (HS; daily arithmetic averages), SWE (daily maxima), and GST (daily arithmetic averages). For a quarterly and yearly comparison, descriptive statistics as well as number of frost days, icing days, 5 °C-days, 10 °C-days, and 15 °C-days, and rain days were computed (Tab. S1). Malfunctioning sensors caused data gaps during two periods: from 01 January to 31 August 2020 (SWE; IMIS station BER3) and from 01 May to 12 August 2022 (RA; IMIS station BER2).

## 2.3 Ground temperature and piezometric pressure data

Ground temperature was measured hourly in boreholes B1 and B5 on the Ursina III rock glacier (Fig. 1 c – f). Borehole B1 was drilled to a depth of 67 m in 1990 (Vonder Mühll and Holub, 1992). In 2005, after the original thermistor chains were sheared off, the borehole was instrumented with 16 YSI thermistors of type 44006 with an accuracy of ± 0.1 °C (YSI Inc., Yellow Springs, OH, USA, www.ysi.com) to a depth of 15.2 m (Fig. 2). B1 temperature data is accessible via the PERMOS network, www.permos.ch (Permos, 2023a). Borehole B5 is equipped with ten Keller PAA-36XiW piezometers (accuracy ± 0.115 bar) comprising a PT1000 temperature sensor (accuracy ± 0.01 °C), measuring the effective pressure at the sensor's membrane relative to vacuum (KELLER Druckmesstechnik AG, Winterthur, Switzerland, http://keller-druck.com; Fig. 2). For more details on the drillings and sensors see Phillips et al. (2023).

We calculated the annual arithmetic temperature averages for each sensor in borehole B1 from 2014 to 2022 to show the annual temperature variations. For B5, we computed daily arithmetic averages for each sensor to depict the short-term evolution of ground temperature and piezometric pressure between 01 January 2021 and 01 June 2023.

## 2.4 Cross-borehole electrical resistivity tomography

We use cross-borehole time-lapse ERT to monitor and visualize resistivity changes and thus gain insight into changes in the ice / water ratio of the Ursina III rock glacier. Cross-borehole ERT measurements were conducted on the 20th of each month from August 2021 to June 2023 (in total 23 time steps). Data for 20 March 2023 are not available due to a system malfunction. We, therefore, use data from 29 March 2023. Data was acquired using a permanently installed cross-borehole ERT setup (Phillips et al., 2023) consisting of 48 electrodes with 0.5 m electrode spacing installed in parallel in boreholes B3 and B4 (Figs. 1 c – f and 2). We used stainless steel electrodes (l = 100 mm; d = 13 mm) integrated on a multi-core ERT cable. To improve contact with the ground, we installed stainless steel clamp collars (h = 11 mm; D = 34 mm). To establish contact between the electrodes and the walls of the boreholes, we filled the boreholes with a mixture of sand (grain size ≤ 2 mm) and gravel (> 2 – ≤ 4 mm) in a ratio of 1:1. The two boreholes are 5 m apart, and the two end electrodes are at a depth of 11.5 m, reaching the top of the shear horizon, which was determined by Arenson et al. (2002). We used a Syscal Pro Switch 48 resistivity meter and a Syscal Monitoring Unit to automatically collect, record, and transmit the acquired data (IRIS Instruments, Orléans, France, www.iris-intruments.com). 1494 direct and reciprocal data points were collected for each ERT time step with a dipole-dipole skip-two configuration (dipole spacing of three electrodes).

ERT data was processed using the Python-based open-source software ResIPy Vers. 3.4.5 (Blanchy et al., 2020). ResIPy uses the mature R2 code to invert ERT data. Only (i) paired data, i.e. data with a direct and a reciprocal measurement, (ii) positive apparent electrical resistivities, and (iii) data with a reciprocal error of less than 25 % were used for the inversion. As the injected current flows vertically and laterally, a triangular mesh was created that extended the area of interest by 10 m laterally and downwards (Phillips et al., 2023), with progressively coarser elements at increasing distances from the target area. ResIPy's time-lapse algorithm is based on the difference inversion method of Labrecque and Yang (2001). Based on Occam's inverse method (Binley and Slater, 2020), the background data were inverted in the first step. Subsequent data sets subtracted the background data before inverting the data with the difference algorithm, which attempts to reduce the misfit between the difference in two data sets and the difference between two model responses (Labrecque and Yang, 2001).  According to the reciprocal error check (Binley, 2015), fitting a power-law error model, and evaluating the normalized inversion errors (Blanchy et al., 2020), an expected data error of 10% was defined for the inversion process. The 20 August 2021 model (ERT01) is used as a reference (background model), i.e. changes in resistivity are expressed as a percentage difference from this first reference survey. The inversion converged after five iterations (final RMS misfit: 1.09; remaining data points: 355). All other models (ERT02 – ERT23) converged after a maximum of two iterations (average of remaining data points after filtering: 457; range:

242 – 635). Tomograms are presented as absolute values of electrical resistivity on a logarithmic scale (log10) and as images showing the percentage changes in resistivity along the background distribution.

## 2.5 Terrestrial laser scanning

We performed terrestrial laser scans (TLS) with a VZ6000 Riegl long-range scanner (RIEGL Laser Measurement Systems GmbH, Horn, Austria, www.riegl.com) to determine year-to-year horizontal surface displacement rates of the Ursina III rock glacier lobe. The detailed procedure is described in Kenner et al. (2020). It includes (i) acquisition of the point cloud with a minimum resolution of 10 cm, (ii) generation of a digital elevation model with a subsequent high-pass filter, and (iii) application of the 2D feature tracking algorithm "particle imaging velocimetry" by Roesgen and Totaro (1995). Surface displacement values in stable areas serve as a rough estimate of the accuracy of the method and allow for the correction of a systematic error, resulting in error values distributed around zero. Data were collected yearly after snow melt in July. We show data from the period 2018/19 onwards.

## 2.6 Data visualization and statistical analysis

We used violin plots (Fig. 8) to compare ground temperature, piezometric pressure, and electrical resistivity over time (Hintze and Nelson, 1998). The relevant data are presented for the 20th of every month between January 2021 and June 2023, except for March 2023, where data for 29 March are used as there were no data available for 20 March. The violin plots show the continuous distribution of the three variables on the respective day over the depth range of the ground temperature sensors (2.0 – 8.5 m) and the piezometric pressure sensors (4.0 – 8.5 m). Resistivities were extracted from the inverted resistivity models between 2.0 – 8.5 m. Here, the violins show the mirrored density distribution of the data by using a Gaussian kernel smoother and a bandwidth corresponding to the standard deviation of the kernel. In the centre of the violins are the notched boxplots. The notches provide a 95 % confidence interval to compare the medians (Mcgill et al., 1978). Additionally, arithmetic averages were calculated.

The robust non-parametric Kruskal-Wallis H-test statistic was applied to determine year-to-year differences of the variables ground temperature, piezometric pressure, and electrical resistivity per month (Kruskal and Wallis, 1952). Based on the H-test, the eta-squared measure $\eta H2$ was computed to assess the strength of the statistical relation. The effect size estimate indicates either (i) a small ($\eta H2$: 0.01 to < 0.06), (ii) a moderate ($\eta H2$: 0.06 to < 0.14), or (iii) a large effect ($\eta H2$: $\geq$ 0.14) (Tomczak and Tomczak, 2014). A significant Kruskal-Wallis test requires a subsequent pairwise comparison to determine which groups differ. To do this, a Dunn's test was conducted with a highly conservative Bonferroni correction (Dunn, 1964). We analyzed the horizontal rock glacier surface displacement rates in the same manner (Fig. 8). Violin plots show the year-to-year horizontal displacement velocities, and the described test statistic was performed to assess year-to-year differences.

ERT tomograms were created with the open-source and Python-based ParaView (V 5.11.2) visualization software (Ayachit, 2015), along with freely available scientifically derived colour maps (Crameri, 2023; Crameri et al., 2020). We processed,

analyzed and plotted all other data with R (R-Core-Team, 2022) within the R studio environment (Posit-Team, 2022). We

performed the test statistics with the rstatix R-package (Kassambara, 2023).

## 3 Results

### 3.1 Annual and seasonal patterns of climate variables

The meteorological datasets of the IMIS stations BER2 and BER3 show a similar pattern over time between January 2020 and June 2023 (Figs. 3 and S1, Tab. S2). The air temperature (TA) was highest during the summer heat wave in 2022, when more

205   days exceeded 10 °C and 15 °C than in the other two summers. TA were lowest in winter 2021/2022. Rainfall (RA) varied between the stations (Figs. 3 and S1, Tab. S2). The highest amounts of RA were recorded in 2020. In 2020, there was more intense rainfall based on the daily RA sums and the number of RA days. To compare with long-term climate data from the region see Fig. 10.

Snow cover was thickest with correspondingly high snow water equivalent (SWE) in winter 2020/2021 (Figs. 3 and

S1, Tab. S2). The winters 2021/22 and 2022/23 had a two-stage snow cover build-up, with snow-poor autumns. Snow cover duration was longest in winter 2020/21 and shortest in 2021/22. In 2020, during snow melt, the spring zero curtain (SZC) started earliest. In 2021, the start of the (SZC) was delayed by approximately three weeks compared to the other years and the SZC persisted longest (Fig. 3e, Tab. S3). The shortest SZC period occurred in 2022. Maximum ground surface temperature (GST) was recorded in summer 2022 and highest values were already registered in July (heat wave). The lowest GST was

recorded in winter 2022/23.

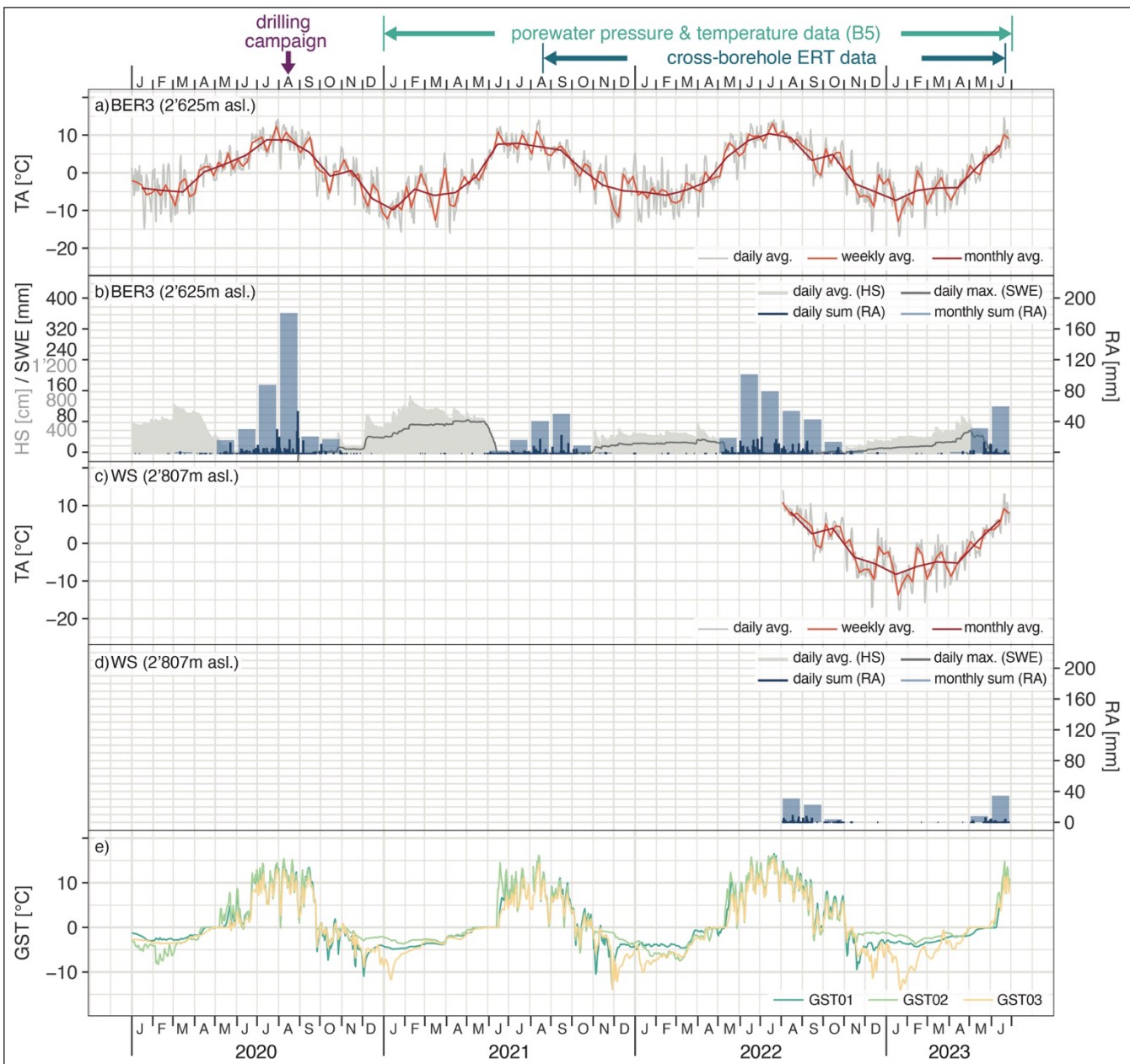

**Figure 3: Air temperature (TA), snow height (HS), modelled snow water equivalent (SWE), and rainfall (RA) at the IMIS station Puoz Bass (BER3) (a and b), air temperature (TA) and rainfall (RA) from the Schafberg weather station (WS) (c and d), and ground surface temperature (GST) for the three miniature temperature data loggers GST01, GST02, and GST03 (e), for the observation period January 2020 to June 2023. Legends are embedded in each individual diagram. Boreholes B3, B4 and B5 were drilled in August 2020 (purple arrow above). The observation periods for pore water pressure, temperature records, and electrical resistivity tomography (ERT) time steps are marked with mint green and light blue arrows, respectively. WS has recorded data since 05 August 2022. Note the data gap for SWE between 01 January 2020 and 31 August 2020 (b). See Fig. 1d and Tab. S1 for location and detailed station information, Tab. S2 for quarterly and yearly metrics, and Fig. S1 for the IMIS station BER2.**

## 3.2 Evolution of ground temperature and pore water pressure

Between 2014 and 2022, the vertical mean annual ground temperature (MAGT) profiles in borehole B1 showed the lowest values in 2016 and 2017 (Fig. 4). The highest temperature values in the active layer ($\sim$ 4 m; AL) were recorded in 2022. However, during the 2022 heat wave, MAGT were lower in the underlying permafrost than in 2020 and 2021. The highest MAGT in the complete profile below 4 m was registered in 2021.

In borehole B5, the active layer thickness (ALT) remained at $\sim$ 4 m during both summers 2021 and 2022 (Fig. 5a). Summer 2022 had a longer active layer duration (155 days) than 2021 (105 days), with higher active layer temperatures. A comparison of the freezing periods revealed that ground temperature values were lowest in the first half of 2022. In winter, the $-1$ °C isotherm extended to 6 m in 2021, below 8.5 m in 2022, and to 8.5 m in 2023. Average daily ground temperatures below $-1$ °C were persistent for over 113 days in 2021, 145 days in 2022, and 130 days in 2023.

The piezometric pressure data shows a five-layer structure with decreasing values from January 2021 to June 2023 (Fig. 5b). The uppermost $\sim$ 4 m (AL) had the lowest piezometric pressures over the whole period. The first noticeable fluctuations occurred at around 4 m, where pressures increased in spring, peaked in summer, and then dropped to a winter low. At $\sim$ 6 m depth, a discontinuous band of anomalies with higher pressure (1.5 bar isobars) was observed. Below this is a zone with lower pressures, decreasing from August 2021 onwards. At a depth of $\sim$ 7.5 m a low-pressure anomaly formed in winter 2023 (< 1.0 bar). At the bottom of the profile there was a band with higher piezometric pressure which decreased over time.

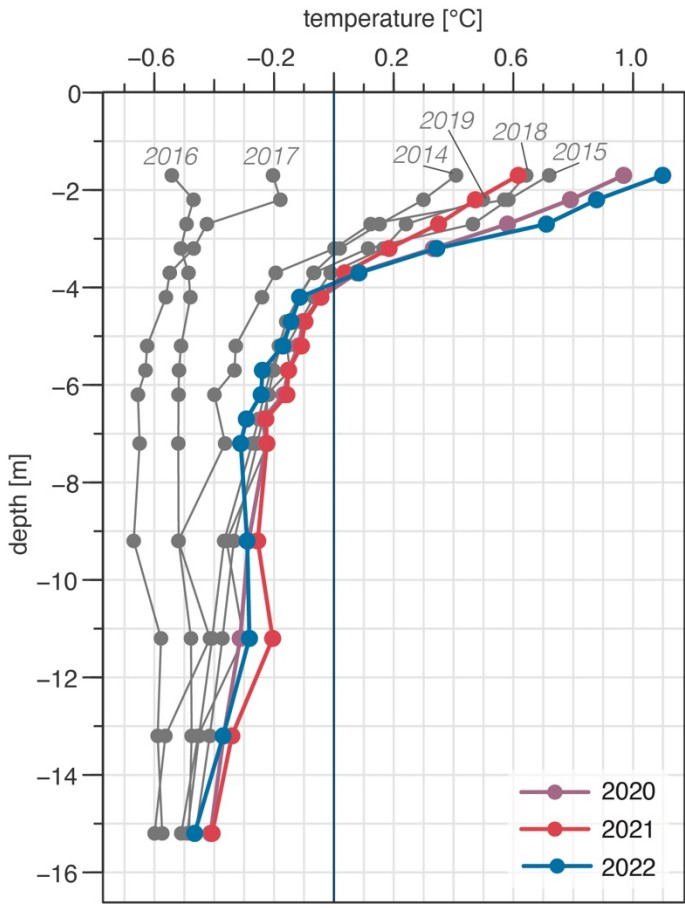

**Figure 4: Vertical temperature profiles of mean annual ground temperature in borehole B1 from 2014 to 2022. The dark blue vertical line denotes the 0 °C-isotherm. Filled circles mark the depths of the thermistors with available data. The years of the observation period have been highlighted (2020 – purple; 2021 – red; 2022 – blue). Note that data for 2023 are not yet available. All other years are shown in grey with the corresponding year label at the top of the profile. See Fig. 1 (d – f) for borehole location.**

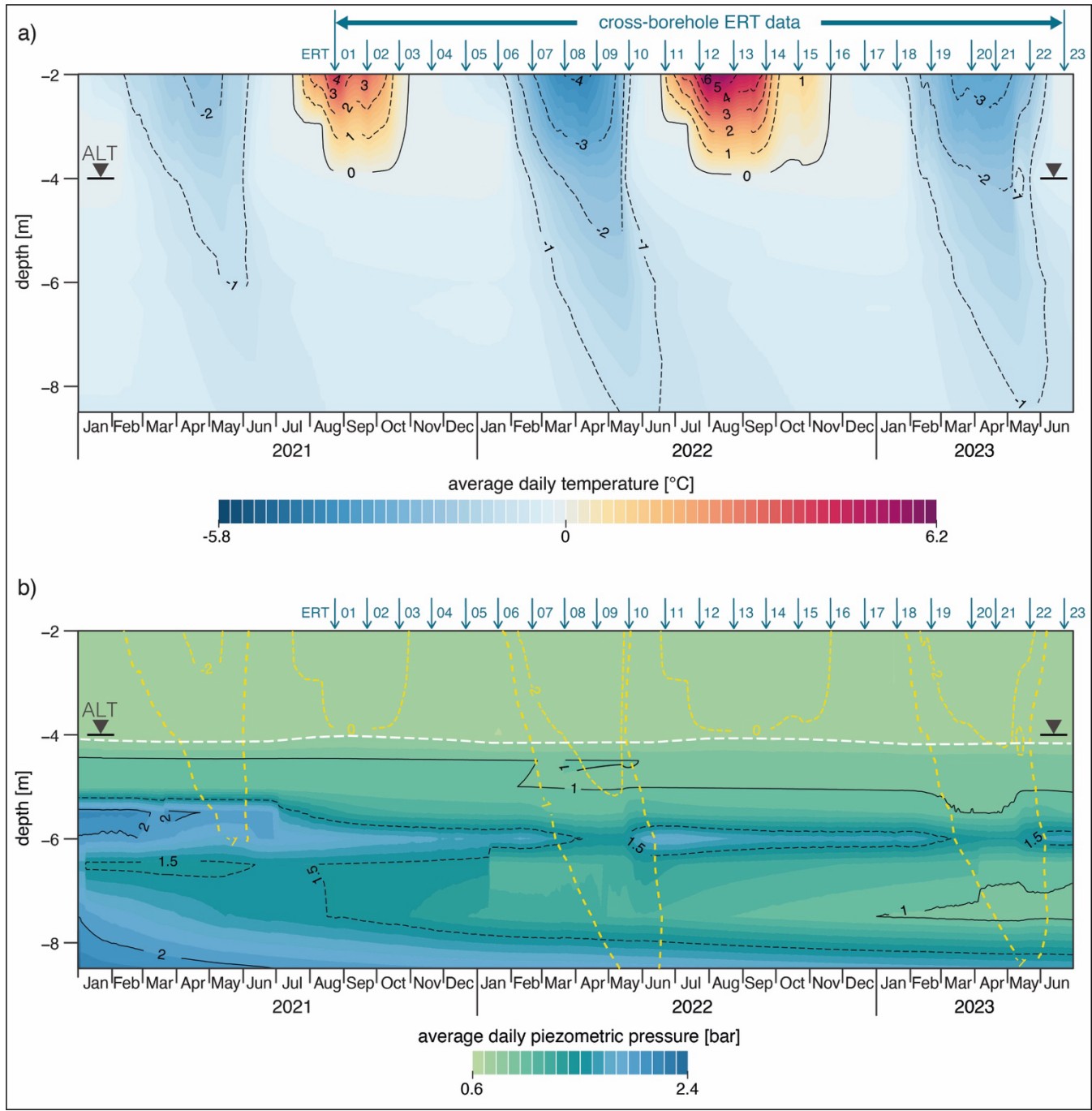

**Figure 5: Time-depth-temperature (a) and time-depth-piezometric pressure (b) plots for borehole B5 from January 2021 to June 2023. Hourly data were aggerated to daily values before plotting (arithmetic average). In addition to the colouring, integer isotherms and selected isobars are highlighted with solid (0 °C, 1 and 2 bar, respectively) and dashed black lines. Thermistors and piezometers are located at depths of 2, 3, 4, 4.5, 5, 5.5, 6, 6.5, 7.5, and 8.5 m. The black triangle indicates the maximum thickness of the active layer (ALT). Fig. 5b additionally highlights the varying piezometric pressure at the ALT (white dashed line shows the depth of thaw), and shows 0 °C, -1 °C and -2 °C isotherms (yellow dashed lines). See Fig. 1 for borehole location, and Fig. 2 for stratigraphy. The**

**top two metres are blanked out as there are no sensors. The time of the 23 cross-borehole ERT time steps analysed is shown in blue at the top of each contour plot (20th of each month; except 29 March 2023 / ERT20).**

### 3.3 Monthly development of cross-borehole electrical resistivity data

The ERT01 background resistivity model represented the summer conditions one year after drilling (Figs. 6 and 7). ERT01 revealed a distinctive zone with low resistivities from the surface of the left hand side to ~ 10 m depth on the right hand side (< 13 kΩm; orange-brown colors). The active layer had a low (~ 1 kΩm; top left) and high (~ < 100 kΩm, bottom right) resistivity segment. Below the AL, a smaller anomaly existed (left side in the tomogram). The lowest resistivities (~ 2 kΩm) were observed at 7 – 8 m depth (right). Another low resistivity area (~ 6.5 kΩm) was located at 7 – 10 m (left). High resistivity areas appeared below the AL at ~ 4 m (left; up to ~ 70 kΩm), below ~ 3 m depth (right; > 80 kΩm) and at ~ 10 m depth (> 80 kΩm). This general pattern was consistent across all the tomograms (ERT01 to ERT23), albeit with varying resistivity values.

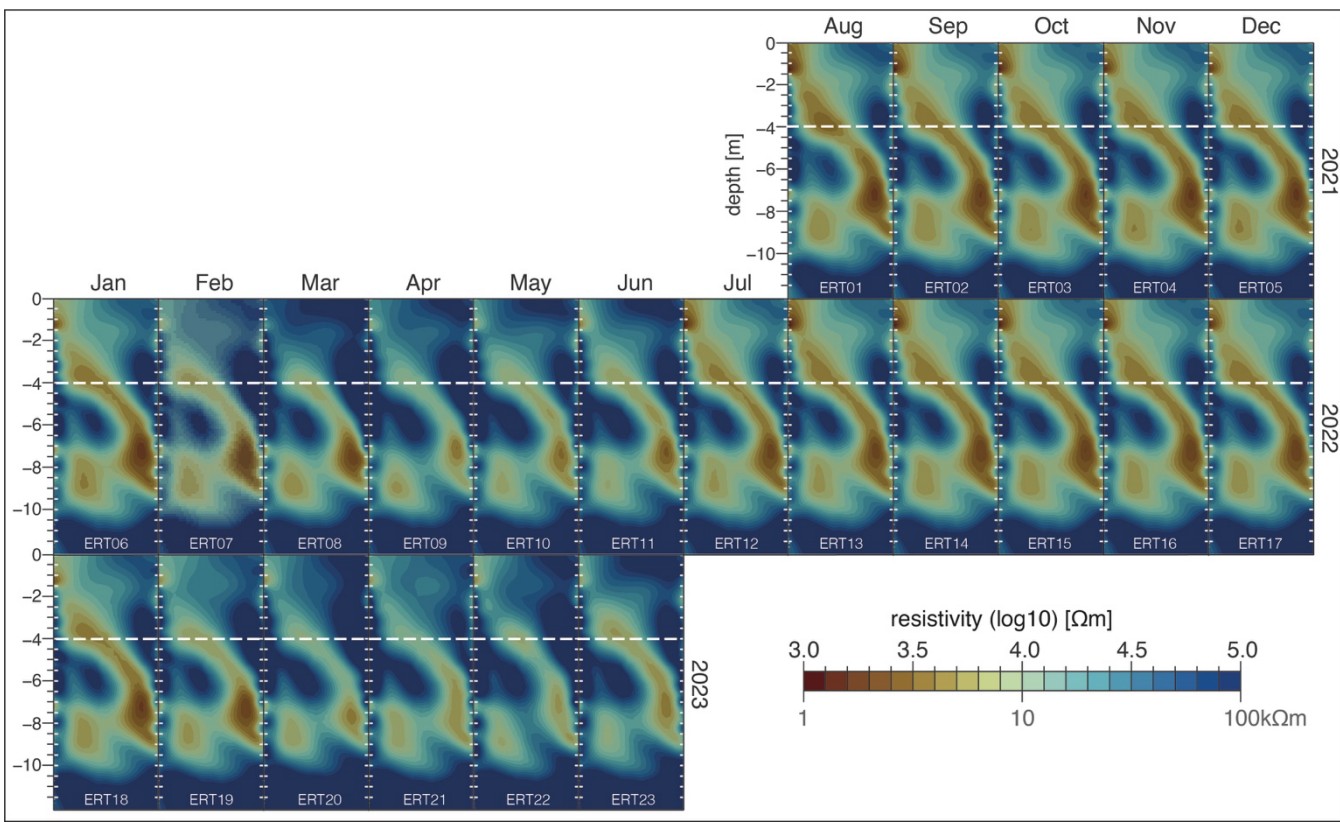

**Figure 6: Inverted resistivity tomograms of the 23 selected cross-borehole ERT datasets between August 2021 and June 2023. The data behind each tomogram were recorded on the 20th of each month (except 29 March 2023). Years are shown on the right of the figure, months on the top. The naming of the tomograms follows the date in ascending order and is indicated at the bottom of the tomogram in white light letters (ERT01 to ERT23). Electrode positions are indicated by the light grey short lines. Borehole B4 is located to the right of each tomogram, B3 to its left. The white dashed lines indicate the maximum depth of the active layer.**

The most prominent resistivity changes relative to ERT01, were observed during late winter and spring (Fig. 7; ERT07 to ERT11; ERT19 to ERT23). In the upper 4 m, resistivities increased substantially, often exceeding 200 % difference. These changes were already noticeable during autumn and early winter (ERT04 to ERT06 and ERT16 to ERT18). Likewise, resistivity increases at ~ 10 m depth were already apparent in the autumn and early winter months, with the increase being more pronounced in autumn 2022 and even more so in the late winter and spring months of 2022 and 2023. Further changes appeared in the zone of the low resistivity anomaly (~ 7 m), with the highest changes (> 200%) in April/May 2022 and March/May 2023.

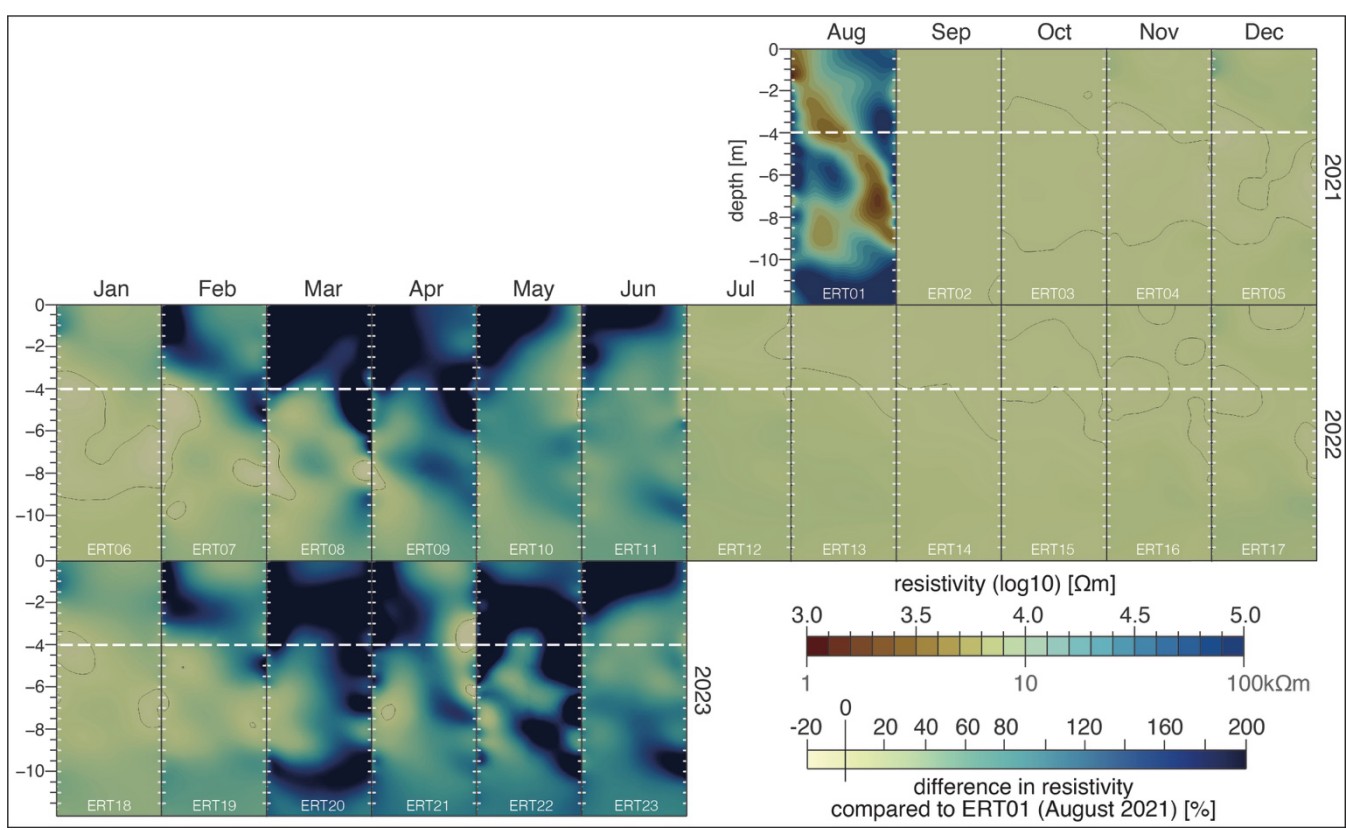

**Figure 7: Relative changes in inverted electrical resistivity based on the first ERT dataset on 20 August 2021 (ERT01) between September 2021 and June 2023. The data behind each tomogram were recorded on the 20th of each month (except 29 March 2023). Years are shown on the right of the figure, months on the top. The naming of the tomograms follows the date in ascending order and is indicated at the bottom of the tomogram in white light letters (ERT01 to ERT23). Electrode positions are indicated by the light grey short lines. Borehole B4 is located to the right of each tomogram, B3 to its left. The thin solid lines denote the 0 % isoline. The white dashed lines indicate the maximum depth of the active layer.**

**3.4 Comparison of ground temperature, piezometric pressures and resistivities**

The violin plots facilitate year-to-year comparisons of ground temperature, piezometric pressure, and inverted resistivity on a monthly basis, including statistical differences (Fig. 8, Tabs. S4 and S5). The Kruskal-Wallis test consistently yielded significant p-values for all three variables, with moderate effect sizes for piezometric pressure and resistivity and a large effect size for ground temperature (Tab. S4).

In 2021, median ground temperatures were consistently higher than in 2022 and 2023, with significant differences in late winter (Fig. 8a). Ground temperatures were slightly higher between February and April in 2023 than in 2022. The median piezometric pressure in 2021 consistently exceeded that of the other years, with significant differences in the winter and spring months (Fig. 8b). In 2022, the median piezometric pressure was higher than in 2023, again with significant differences in late winter and spring. Inverted resistivities showed a less clear signal (Fig. 8c). Generally, resistivity values during winter and spring 2023 were above those in 2022, except for April. Significant resistivity increases were observed in April and May.

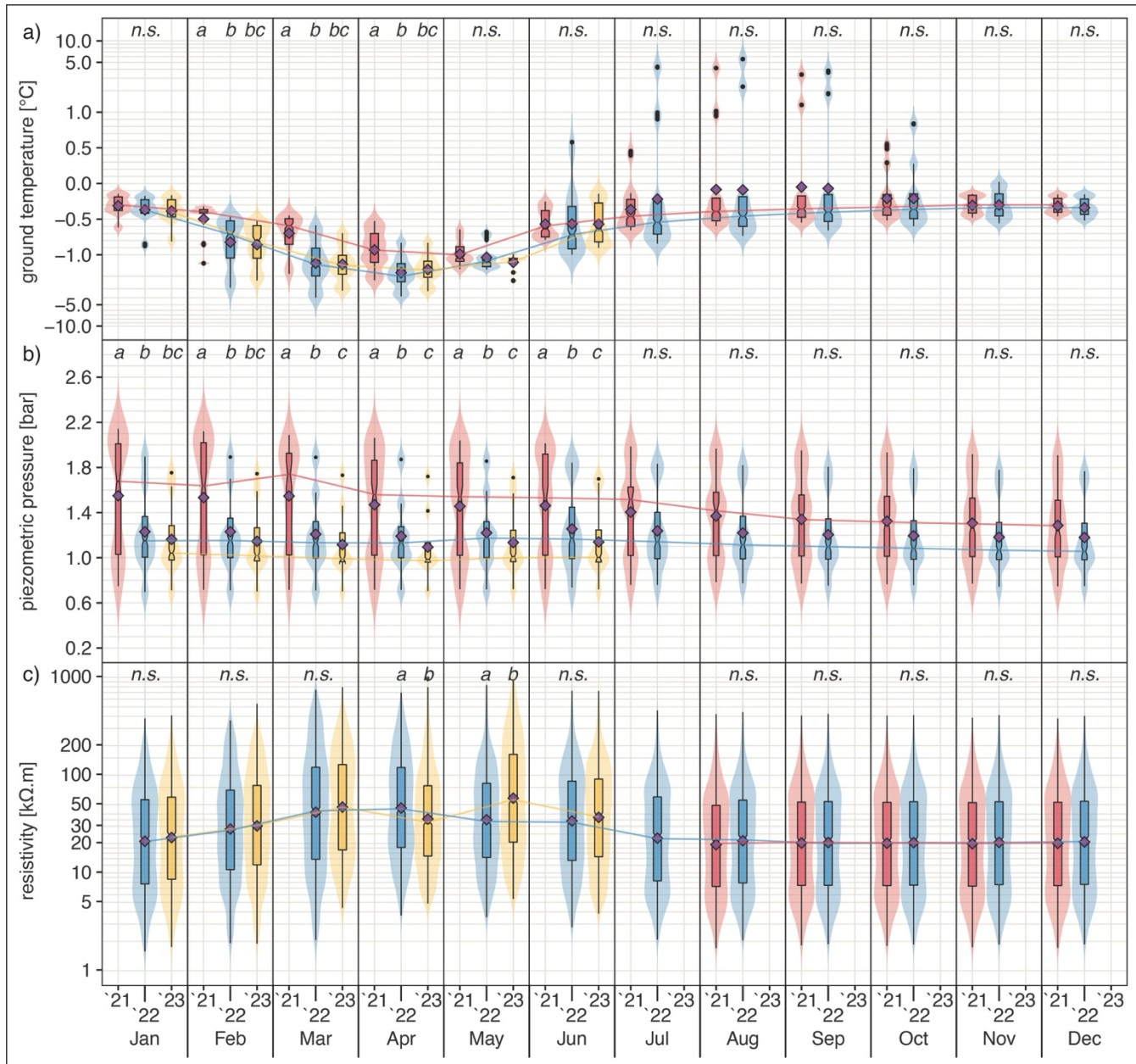

Figure 8: Violin plots of temperature (a), piezometric pressure (b) and inverted resistivities (c) for the 20th / 29th of each month with available data in 2021 (red), 2022 (blue) and 2023 (yellow) – see also Fig. 4. The shaded areas represent the density distribution (violins), with the corresponding notched box plots and related outliers (black dots). The solid lines (red 2021, blue 2022 and yellow 2023) join the median values. Arithmetic averages are represented by purple diamonds. Year-to-year significance levels based on a Kruskal-Wallis *H*-test statistic and Dunn's pairwise comparison test (lower case italic letters) are shown at the top of each diagram (*n.s.*: not significant; violins with a different lowercase letter show a statistically significant difference, P ≤ 0.5). Note that the y axes of a) and c) are symlog transformed.

## 3.5 Annual horizontal surface displacement rates

Between 2019 and 2023, horizontal surface displacement rates varied strongly (Fig. 9, Tabs. S4 and S5). The Kruskal-Wallis
test revealed a significant interannual difference with a large effect size. In the period 2020/2021, rock glacier displacement
velocities were highest and then decelerated in the periods 2021/2022 and 2022/2023, when they reached their lowest rate.
The changes are statistically significant.

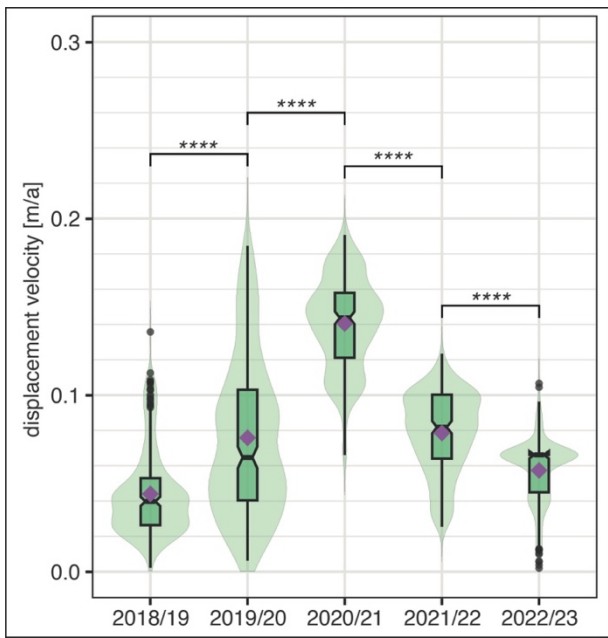

**Figure 9: Violin plots for the year-to-year horizontal displacement velocities of the Ursina III rock glacier lobe based on terrestrial**
**laser scanning (TLS). TLS data were collected in July of each year from 2018 to 2023. The shaded green areas represent the density**
**distribution (violins), with the corresponding notched box plots and related outliers (black dots). Arithmetic averages are**
**represented by purple diamonds. Year-to-year significance levels based on a Kruskal-Wallis *H*-test statistic and Dunn's pairwise**
**comparison test are shown at the top (turned squared brackets; *n.s.*: not significant; **** significant difference, P ≤ 0.0001). Note**
**that velocities for the 2019 reference year 2018 are not shown and were not included in the inferential statistics.**

## 4 Discussion

### 4.1 Towards a generic rock glacier model to explain rock glacier kinematics: rock glacier deceleration

Using annual terrestrial laser scans (TLS), we found that the horizontal displacement velocity of the rock glacier was highest
in the period  2020/2021 (Fig. 9), the highest recorded since the start of the TLS series in 2009 (Kenner et al., 2020, Fig. 4a).
The velocities decreased significantly in the following two measurement periods. This result agrees with findings on other
rock glaciers in the Swiss Alps within the same observation period (2018 – 2023), which exhibited an average velocity decline
of -34 % (Permos, 2023b).

The analysis of nearby IMIS weather station data showed that the summers 2020 and 2021 were cooler, and particularly
summer 2020 was wetter than the dry and hot summer 2022 (Fig. 3, Fig. S1, Tab. S2). Both winters 2021/22 and 2022/23 had very little precipitation compared to the snow rich winter 2020/2021. The temperature and precipitation deviations from the 1991-2020 norm at the nearby weather station on Piz Corvatsch (located at 3297 m asl, 12 km SW of the Ursina III rock glacier) agree with the observations from the IMIS stations, confirming the wetter summer 2020, and the pronounced summer heat wave of 2022 followed by two winters with little snowfall (Fig. 10).

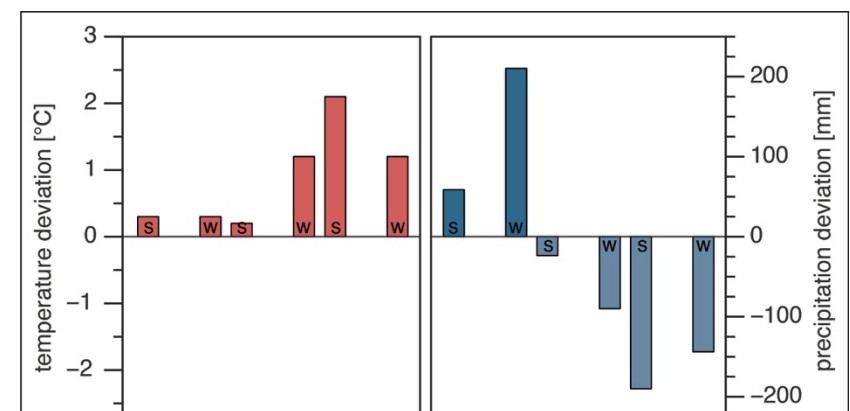

**Figure 10: Changes in air temperature TA (a) and precipitation PRE (b) for the MeteoSwiss weather station at Piz Corvatsch (3297 m asl, 46.42° N / 9.82° E) in relation to the reference period 1991 - 2020. The temperature values are given as mean values, the precipitation values as sums, both for the period 2020 – 2023. Bars with _s_ indicate the summer months (June, July, August), and**
340 **with _w_ winter months (December, January, February). Positive temperature changes are in red (warmer), positive precipitation changes are in dark blue (wetter), negative precipitation changes are in light blue (drier). Norm values: $T_{JJA} = 2$ °C; $T_{DJF} = - 11$ °C; $PRE_{JJA} = 413$ mm; $PRE_{DJF} = 234$ mm. Data source: Meteoswiss (2024).**

Since the start of the measurements in 2014, the active layer in borehole B1 was warmest in 2022 (Fig. 4). The 2022 heat wave
similarly affected active layer duration and -temperature in borehole B5 (Fig. 5a). In parallel, we observed a significant deceleration of the rock glacier in the two periods 2021/2022 and 2022/2023 (Fig. 9).

In the blocky active layer (ALT ~ 4 m), piezometric pressure followed barometric pressure (Fig. 5b) as the sensors hanging between the blocks did not encounter standing water and snow melt water and rainfall flowed past them. At the base of the
350 active layer, the sensors recorded the onset of snowmelt (Fig. 3e, Tab. S3), indicating seasonal water content fluctuations and water accumulation at the permafrost table (dashed white line, Fig. 5b). Pore water pressure increased with the onset of the spring zero curtain (SZC) until summer, confirming previous observations of quick snowmelt and rainwater percolation through the course blocks of the active layer (Ikeda et al., 2008; Krainer and Mostler, 2002). Hence, our data indicates a suprapermafrost water flux along the permafrost table from the onset of snowmelt until late summer/autumn (Fig. 5b and 6).

At around 6 m depth, elevated pore water pressures occurred between permafrost ice and mud-ice layers (Fig. 2 and 5) with the highest piezometric pressures recorded at the beginning of the time series in winter/spring 2021. These then decreased in 2022 and 2023, due to snow-poor winters allowing the ground to freeze more efficiently (yellow-dashed isotherms in Fig. 5) indicating a lower ground water content. At ~ 7 m depth, lower pore water pressure values were registered. The high pressure values in 2021 were linked to the snow-rich winter 2020/21, with a significantly higher snow water equivalent (SWE) and longer snowmelt period compared to the subsequent winters (Fig. 3 and 5b, Fig. S1, Tabs. S2 and S3), and to higher summer rainfall in 2020 and 2021. Ground freezing impacted piezometric pressures in ~ 7 m depth during the late winter/spring months in 2022 and 2023, causing lower pore water pressure values. At ~ 8 m depth, pore water pressure gradually decreased throughout the measurement period due to a decreasing pore water pressure.

The monthly cross-borehole ERT data revealed relative changes in electrical resistivity related to phase changes in water and/or ice content across a 5 x 11.5 m area, complementing the piezometer data (Fig. 6 and 7). Despite uncertainties in the contact between electrodes and ground material (Phillips et al., 2023), the consistent structure in all tomograms and the similarity to the August 2020 borehole stratigraphies (Phillips et al. 2023) across the summer images promotes confidence in data reliability. Although uncertainties remain (see chapter 4.2), our modelled resistivities are consistent with recent surface geophysical measurements near B3 and B4 (Boaga et al., 2020; Pavoni et al., 2023) and with the highly heterogeneous nature of the ground as demonstrated by several drillings within close proximity.

Lower resistivities were observed in the thawed active layer next to a high resistivity anomaly indicating the uppermost blocky material but also a certain heterogeneity within the active layer (Fig. 6). The relative resistivity changes illustrate the freezing and thawing of the active layer and are in line with the ground temperature and piezometric pressure data (Fig. 7). At the base of the active layer, the relatively low resistivity anomaly at ~ 4 m indicates a slightly higher water content in the fines/coarse sediments with ice from the time of the snow melt until early winter and indicates the presence of suprapermafrost water as was also revealed by the piezometric data. In deeper ice-bearing layers (~ 4 to ~ 6 m, and from ~ 9 down), resistivities were highest through all seasons. The low resistivities at ~ 7 m depth are remarkable. Despite ground temperatures below 0 °C (Figs. 4 and 5a), resistivities of ~ 2 kΩm (summer tomograms) indicate a considerable amount of unfrozen water, as was confirmed by the presence of sludge containing ice crystals during drilling in 2020. The presence of unfrozen water, i.e. liquid water in frozen ground at temperatures below the phase equilibrium temperature (Romanovsky and Osterkamp, 2000), due to pressure conditions, grain size, saline water and/or other soil properties, has been quantified in several laboratory (e.g., Williams, 1964), field experiments (e.g., Oldenborger and Leblanc, 2018) and in modelling studies (e.g., Bi et al., 2023). However, less is known about the amount of unfrozen water in coarse-grained ground, particularly in rock glaciers, and our data suggest intrapermafrost water fluxes. From summer until February in both 2022 and 2023, only little seasonal resistivity changes were registered in these wet sludge layers, which might be explained by latent heat effects (Phillips et al., 2023). However, with the subsequent temperature decrease between January and June and the resultant phase change, the resistivity increased. This effect was greater in 2023 than in 2022, reflecting the drier ground conditions in 2023, as is also shown by the

piezometric pressure data. The relative resistivity changes in the tomograms below ~ 9 m were prominent. The ERT time-lapse images emphasize the drier conditions in 2023 compared to 2022. This observation agrees with the piezometer data, which are available to 8.5 m depth but is only fully captured in the ERT data down to ~ 12 m, providing a more comprehensive understanding of the conditions at the top of the shear horizon.

Statistical analyses of ground temperature, piezometric pressure and electrical resistivity data clearly underline the relations described above and emphasize the crucial role of ground cooling and drying on rock glacier deceleration, which is confirmed by the statistical significances in the winter and spring months (Fig. 8).

Recent studies argue that the primary factor influencing rock glacier deformation variations is water from liquid precipitation and snow melt (e.g., Wirz et al., 2016; Cicoira et al., 2019a). Our study indicates that snow cover acts as the catalyst influencing rock glacier velocity via ground temperature, phase change, and water input during snowmelt. However, we could not assess the impact of rainfall as the analyzed period mainly experienced low or light rainfall (Fig. 3, Fig. S1, Tab. S2). The higher pore water pressure in early 2021 likely resulted from the wetter 2020 summer and autumn, characterized by higher rainfall intensities (Fig. 3 and 5b, Fig. S1, Tab. S2). This suggests that rock glacier deceleration requires winters with little snow and relatively dry summers.

Like other rock glaciers, Ursina III has temperatures near the melting point of ice (Figs. 4 and 5a) (Permos, 2023b) implying that liquid water coexists with ice, significantly influencing the viscosity of the permafrost ice (Ikeda et al., 2008; Arenson et al., 2002). The piezometric pressure and ERT data confirm that rock glaciers at or near their melting point can contain a substantial amount of unfrozen water in the permafrost. The stratigraphic recordings and ERT data represent the small-scale heterogeneity within the rock glacier, i.e. the low resistivity anomalies throughout the ERT images indicating a considerable amount of unfrozen water at around 7 m depth. This suggests there is the potential for water to flow from the active layer into deeper layers via preferential pathways, that is intrapermafrost water flow. This is consistent with other studies where water movement has been found to create high structural heterogeneity and favors flow paths within the ice-rich zone (Krainer and Mostler, 2002; Cicoira et al., 2021; Zenklusen Mutter and Phillips, 2012a).

From 1991 to 2000, the shear zone in borehole B1 on rock glacier Ursina III was in a depth between 16.4 m and 11.4 m, with minimal deformation above (Arenson et al., 2002), signifying that most of the rock glacier deformation occurred within the shear horizon. Previous studies suggested that increased water infiltration raises pore water pressure within the rock glacier (Cicoira et al., 2019b; Kenner et al., 2017; Ikeda et al., 2008), subsequently reducing shear strength and enhancing deformation. However, the depth of water infiltration has not previously been determined. Considering that the shear horizon is a relatively thin layer of a few meters thickness (Cicoira et al., 2021) our piezometer and ERT data provide insight into the upper reaches of the shear horizon. In 2021, high pore water pressure, linked to a wet 2020 summer and a thick, insulating snow cover (high

SWE) in the following winter resulted in reduced freezing, higher water contents, and lower shear strengths. This induced increased horizontal deformation rates at the rock glacier surface. Lower pore water pressures in the subsequent years with
425 scant snow cover resulted in the deceleration of the Ursina III rock glacier in 2022 and 2023, as less water infiltrated the shear zone.

## 4.2 ERT data quality and data processing

Conducting ERT measurements in environments with very high resistances and/or low sensitivities is challenging, and surveys require special techniques for sensor coupling, data acquisition and interpretation (Hauck and Kneisel, 2008; Hilbich et al.,
2009; Mollaret et al., 2019) to gain better control over systematic measurement errors that can affect the inversion. To improve electrode coupling and reduce contact resistance, i.e. to increase the surface area of the electrode, we used stainless steel electrodes in combination with steel clamps and used a finer-grained material (sand-gravel mixture) to establish contact between the electrodes and the borehole walls. Poor contact resistances due to alteration effects cannot be controlled by, for example, adding (salt) water or replacing the electrode as can be done for surface measurements. However, to address the
uncertainty of the contact resistance between the electrode and the substrate material, it is possible to record the contact resistances for each ERT measurement. We run the system on a solar panel, with just enough power for a daily measurement and for data transfer. During the winter season the power supply already reaches its limits and data transfer is not always possible. To avoid data loss due to low power, we have, therefore, not recorded contact resistance.

It is important to remove the influence of poor-quality data in order to obtain reliable inversion results despite the possible
high contact resistance (Hilbich et al., 2011). Therefore, we (i) collected reciprocal data and (ii) applied a multi-stage filtering approach to our data before inversion (e.g., Hilbich et al., 2011; Mollaret et al., 2019; Oldenborger and Leblanc, 2018). As charge-up effects can occur at the current electrodes, further contact resistivity errors can arise if these electrodes are subsequently used as potential electrodes (Dahlin, 2000). In our study, we considered measured positive apparent resistivities (Mollaret et al., 2019; Hilbich et al., 2011) but did not investigate possible polarisation effects in detail. However, by collecting
reciprocal data, errors can be better quantified, and the collection of reciprocal data provides a good estimate of the precision of the resistivity data (Oldenborger and Leblanc, 2018; Binley, 2015).

In addition to the possible changes in contact resistance and associated changes in signal strength due to poor electrode coupling or charging effects described above, variations are also caused by seasonal changes in water and/or ice content driven by ground temperature. This means that longer-term repeatability and a changing system over time can lead to higher errors,
which is particularly challenging for time-lapse monitoring systems. Flores Orozco et al. (2019) argue that these temporal variations do not increase the misfit between direct and reciprocal measurements, as these variations are not outliers in the independent datasets and suggest filtering out temporal outliers to avoid systematic errors affecting the temporal variations in the data. In our case, we did not filter for temporal outliers, but due to the scope of our manuscript, we chose the time-lapse inversion approach developed by Labrecque and Yang (2001) to monitor changes in relative ice and/or water content, as their
difference inversion scheme mitigates the effect of systematic errors (Yang et al., 2015).

To our knowledge, the present study is the first to analyse multi-year cross-borehole ERT data in a rock glacier. We used a wide variety of methods to better understand the poorly understood water content in these complex landforms and its influence on their kinematics. Future studies of cross-borehole ERT measurements in permafrost environments should therefore include further quantitative analysis, including synthetic modelling, extended investigations of data filtering and data error estimation, sensitivity analysis, possibly other time-lapse inversion approaches, and keeping the same quadrupoles for each inversion time frame. This will help to better compare inversion results with relatively the same sensitivity across the monitoring domain, and avoid a possible seasonal over- and underfitting.

## 5 Conclusions

We investigated the conditions leading to the deceleration of the Schafberg Ursina III rock glacier between 2021 and 2023, a period including the summer 2022 heat wave. We used a novel combination of borehole temperature, piezometric pressure, cross-borehole ERT, and TLS data together with meteorological data from nearby weather stations. Our monitoring approach allowed us to observe the relative resistivity changes associated to water content changes of rock glaciers to a depth of ~ 11.5 m on a daily to monthly basis.

Rock glacier deceleration appears to be primarily caused by ground cooling and drying. Winters with little snow lead to decreases in ground water content, as the ground can freeze efficiently in the absence of an insulating snow cover. Low water contents in late winter and early spring are the main driving factor for rock glacier deceleration. Differences in piezometric pressure, ground temperature and ground resistivities were only statistically significant at these times of year.

The summer 2022 heat wave had a negligible effect on the rock glacier temperature, which had cooled efficiently in the preceding winter and temperatures in the permafrost remained low. However, the heat wave affected active layer duration and temperature, which lasted longer than usual and was particularly warm.

Determining the impact of rain on ground temperature and water content in the rock glacier is challenging. Our findings suggest that rainfall did not considerably affect the water content of rock glaciers to depths of ~ 11.5 m over the period considered. We conclude that rainfall must have a certain quantity to be registered in the near subsurface. We did not identify any perennial water accumulation in the active layer, on the permafrost table, nor in the top ~ 11.5 m of the rock glacier during the observation period. However, we registered lateral water flows during snow melt.

To establish a generic process model of rock glacier kinematics and better understand the influence of environmental conditions, future studies must include the influence of liquid precipitation. Similar monitoring systems should be set up on other rock glaciers, including measurements within the shear horizon, which is the layer controlling rock glacier kinematics.

## Code Availability

ERT data were processed with the openly available software ResIPy (Blanchy et al., 2020) and visualized with the open-source visualization software ParaView (Ayachit, 2015), along with freely available scientifically derived colour maps (Crameri, 2023; Crameri et al., 2020). We processed, analyzed and plotted all other data with R (R-Core-Team, 2022) within the R studio environment (Posit-Team, 2022). We performed the test statistics with the *rstatix* R-package (Kassambara, 2023).

## Data Availability

The collected field data, which was used in this article (ground temperature and piezometric pressure data of borehole B5, cross-borehole ERT data, as well as GST data) can be provided by Alexander Bast (alexander.bast@slf.ch) or Marcia Phillips (phillips@slf.ch) on request. Ground temperature data of borehole B1 are available via the Swiss Permafrost Monitoring Network PERMOS (Permos, 2023a). Data for the IMIS snow stations were provided by the IMIS measuring network (Intercantonal and Information System, 2023). Data for the Piz Corvatsch weather station were provided by MeteoSwiss 495 (Meteoswiss, 2024).

## Author Contributions

MP initiated the study, designed the field experiment and was responsible for drilling and borehole instrumentation. MP and AB programmed and maintained the field monitoring equipment. RK carried out the terrestrial laser scans and raw data processing. AB and MP analyzed the data and drafted the manuscript. All authors contributed to the article.

## Competing Interests

The contact author has declared that none of the authors has any competing interests.

## Acknowledgements

We thank Jacopo Boaga and Mirko Pavioni for enlightening discussions and constructive collaboration. Chasper Buchli is thanked for his ongoing dedication during the installation of the devices and subsequent support. We thank the Keller 505 Druckmesstechnik AG team for their valuable support with the piezometers and Helibernina for transport to the research site. We are grateful to the Amt für Naturgefahren AWN Graubünden and the Gemeinde Pontresina for their continuous support. The Schafberg Ursina B1 borehole is part of the PERMOS network. Dimitrios Ntarlagiannis, Andreas Hoerdt, an anonymous reviewer, and the editor, Adrian Flores Orozco, are warmly thanked for their constructive and insightful comments and suggestions.

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
