# Peer review of "Short-term cooling, drying and deceleration of an ice-rich rock glacier"

_EGUsphere, 2024_

## Referee Comment (RC1)

**Short-term cooling, drying and deceleration of an ice-rich rock glacier**

*Bast et al.*

**Summary**

This highly relevant and interesting manuscript presents the combination of borehole temperature, piezometric pressure and geophysical data with surface meteorological and kinematic data to investigate a short-term deceleration of a rock glacier in the Swiss Alps. To the best of my knowledge, the use of borehole ERT combined with borehole temperature and piezometric pressure data in a rock glacier is a novelty and makes the manuscript very interesting.

The results of the study show that the snow cover plays a crucial role for short-term velocity changes due to its effects on the thermal state and particularly due to the water supply. According to this manuscript, snow-poor winters and dry summers are ideal for the short-term deceleration of the rock glacier under investigation. Additionally, the authors demonstrated that a summer heat wave did not play a significant role in the rock glacier movement.

The manuscript is well-written, well-structured and the figures are well designed. The Introduction provides a good overview about the relevance of the study, what has been done so far and the main objectives of the manuscript. In Material and Methods, the authors provided all information needed to understand the data collection, the processing, statistical analyses and visualizations, but a few points in the description of the ERT data processing are missing or should be revised. The results and the discussion are well presented. However, in the discussion the authors could try to focus on explaining links and the (somehow surprising) discrepancy between temperature and resistivity in the permafrost layer. These two issues together with some specific comments and technical corrections (see below) can be easily addressed. Therefore, I recommend accepting this interesting manuscript with minor revisions.

**General Comments**

Processing of electrical resistivity data

- See specific comments below

Discussion of the low resistivity zone in the permafrost layer

- See specific comments below

**Specific comments and technical corrections**

Line 9: „Long term" -> „Long-term" (for consistency)

Line 10: "variability of their velocity and there is still a gap in the" -> "variability of the velocity with a particular gap in the"

Line 11: "role of water in rock glaciers" Role in what? Maybe add here "in rock glacier movement"

Line 14a: "cross-borehole electrical resistivity tomography sensors." -> "cross-borehole electrodes for electrical resistivity tomography measurements"

Line 14b: "using terrestrial" -> "using repeated terrestrial"

Line 17: "lowering of the water content in rock glaciers is crucial for" Where exactly? In general, at the shear horizon, at the surface?

Line 22: "ice/water contents" -> "ice/water content"

Line 23: "better understanding of factors affecting rock glacier kinematics" -> "better understanding of the main drivers of rock glacier kinematics"

Line 26: See also the recently published paper: Kellerer-Pirklbauer, A., Bodin, X., Delaloye, R., Lambiel, C., Gärtner-Roer, I., Bonnefoy-Demongeot, M., Carturan, L., Damm, B., Eulenstein, J., Fischer, A., Hartl, L., Ikeda, A., Kaufmann, V., Krainer, K., Matsuoka, N., Morra Di Cella, U., Noetzli, J., Seppi, R., Scapozza, C., Schoeneich, P., Stocker-Waldhuber, M., Thibert, E., and Zumiani, M.: Acceleration and interannual variability of creep rates in mountain permafrost landforms (rock glacier velocities) in the European Alps in 1995–2022, Environ. Res. Lett., 19, 034022, https://doi.org/10.1088/1748-9326/ad25a4, 2024.

Line 27: "Rock glacier kinematics are driven by common external climatic factors, and they vary over time (Delaloye et al., 2010a; Delaloye et al., 2008), with phases of acceleration, interrupted by periods of stagnation or deceleration of variable duration (Permos, 2023a)." -> "Rock glacier kinematics vary over time (Delaloye et al., 2010a; Delaloye et al., 2008), with phases of acceleration, interrupted by periods of stagnation or deceleration of variable duration (Permos, 2023a), driven by common external climatic factors.

Line 37-40: At this position, the paragraph is a bit lost and disconnected. Can you add it in the first paragraph after the first sentence?

Line 41: Can you specify the type of space-borne data?

Line 47: "rock glacier, and recent literature" -> "rock glacier. Recent literature"

Line 53: "refraction seismic" -> "refraction seismic tomography (SRT)"

Line 55: In the references, maybe you can add: Kneisel, C., Hauck, C., Fortier, R., and Moorman, B.: Advances in geophysical methods for permafrost investigations, Permafrost Periglac., 19, 157–178, https://doi.org/10.1002/ppp.616, 2008.

Line 52-62: Could you try to restructure the paragraph? Maybe you start with the direct method (borehole temperature sensors), their limitations and present the geophysical methods with their advantages over the direct methods as alternatives? What about the spatial coverage/resolution? That would make it clearer why you use a geophysical method in your study.

Line 60-62: "Cross-borehole geophysics are a means to distinguish ice and water in ice-rich permafrost and to monitor their volumetric variation and distribution over time (Musil et al., 2006; Phillips et al., 2023)." Geophysical methods do not provide direct information about ice and water content, you still need a model linking a geophysical variable, like resistivity, to ice or water content. Please add that here.

Line 73: I would remove the apostrophe in the elevation numbers.

Line 77-81: The description of the boreholes and their location is a bit confusing. For example: Are there also boreholes B1 and B2? What is the depth of the boreholes? Did you reach the bedrock in boreholes B4 and B5 as well or just in B3? What  Please try to improve the structure.

Line 83: "relative ice-/water contents" The ERT method does not provide ice-/water contents directly, just a physical parameter.

Fig. 1: I would recommend to merge c and d, remove e, and add the scale bars and symbologies in the accompanying subplot. Additionally, the borehole names B2-B4 are missing. And I guess the north arrow is not valid for all subplots, right?

Fig. 2: "Syscal system" In the caption of the Figure, the type of electrodes would be more relevant than the instrument name.

Line 110: Please add that the Table is in the Supplementary Material.

Line 118: "These delivered" -> "They deliver"

Line 122a: "icing days" -> "number of icing days"

Line 122b: "15 °C days" -> "15 °C-days

Line 126-127 and 130: Please add the information about the boreholes in section 2.1. What is in B1 below 15.2 m? Is it broken?

Line 134-136: Why B1 for the annual variations and B5 for the short-term variations?

Line 135: "the evolution" -> "short-term evolution"

Line 138: The abbreviation ERT has been already explained.

Line 139: "ERT soundings" -> "ERT measurements"

Line 140: "in total 23 ERT measurements" -> "in total 23 time steps"

Line 146a: Please remove the apostrophe in "1494"

Line 146b: "data points" -> "data"

Line 146c: "ERT sounding" -> "time step"

Line 147: Please describe what dipole-dipole and skip-two means.

Line 138-147: please add some information about contact resistances and their change over the whole year. I guess they are not constant over the time and might affect the data quality, particularly during winter months. Which type of electrodes did you use and how are they connected to the subsurface material?

Line 149a: Please explain what reciprocal counterpart means.

Line 149b: "positive values of measured apparent electrical resistivities" -> "positive apparent electrical resistivities"

Line 148-150: How many quadrupoles did you keep for the inversion? Did you filter for every time step the same quadrupoles? If not, the combination of different quadrupoles might result in different sensitivities increasing the generation of time-lapse artifacts.

Line 150-152: 10 m mesh extension is a bit too small and might influence the inversion.

Line 152: "ResIPy's" For inversions, ResIPy calls the R2 code – maybe you can add that here.

Line 154: What means "according to the reciprocal check"? Could you specify the error model here? Is your error value (10%) similar to the one estimated from the normal-reciprocal analysis and is the error constant over all time steps? And why is your used error parameter (10%) much higher than your filtering threshold (25%)?

Line 156a: Could you explain what is the reference model in this context?

Line 156b: "It converged" -> "The inversion converged"

Line 157: Did you use an error-weighted RMS? If yes, you could add that here.

Line 168: "systemic" -> "systematic"

Line 171-179: Could you shift the paragraph with the description of the Figure to the results section?

Line 176: "entire layer" What do you mean with entire layer?

Line 180: "differences of the variables" Which variables?

Fig. 3: The temperature and precipitation data look very similar between the different stations. Could you try to present every parameter only for one station and shift the data of the other stations to the Supplementary material? That would make the statement of the Figure more concise.

Line 211: "(ERT) soundings" -> "(ERT) measurements"

Line 241: "thickness of the active layer (ALT)" -> ""depth of the active layer"

Line 248: "from the surface to ~ 10 m depth" -> "from the surface on the left hand side to ~ 10 m depth at the right hand side"

Line 249a: "low and high resistivity segment" Please add the range of the values.

Line 249b: "Below the AL" Please add here the depth of the AL.

Line 249c: "existed" -> "exists"

Fig. 6: What about the sensitivity in the different time steps and in your model domain? To be more concise, you could merge Fig. 6 with Fig. 7 and keep the absolute values only for the reference time step or one image per season. In the colorbar please write Ωm instead of ohm.m.

Line 256: "cross-borehole electrical resistivity tomography (ERT)" -> "cross-borehole ERT"

Line 259-260: "The ERT multi-core cables with 24 electrodes per borehole (indicated by the light grey short lines) and a vertical spacing of 0.5m are installed the boreholes B3 and B4." You wrote that information already in the Methods. You can change that to: "Electrode positions are indicated by light grey short lines.

Line 262: "See Fig. 1 for borehole locations, Fig. 2 for further details and stratigraphy, and Fig. 7 for relative electrical resistivity changes." Not necessary here.

Line 277-278: "See Fig. 1 for borehole locations, Fig. 2 for further details and stratigraphy, and Fig. 7 for inverted resistivity tomograms." Not necessary here.

Fig. 8: "kohm.m" -> "kΩm"

Line 04: "fastest" -> "highest"

Line 309a: "TLS were collected" -> "TLS data were collected"

Line 309b: "each year" -> "each year from 2018 to 2023"

Line 328: "Changes in temperature T" -> "Changes in air temperature TA"

Fig. 10: In these Figure you use different colors for two different categories which makes it confusing. It would be easier to understand the plot, if you directly write down which column presents winter and which one summer and you keep the bars in the same color.

Line 343: I think "SZC" has not been introduced before.

Line 355a: "electrical resistivity tomography (ERT)" -> "ERT"

Line 355b: "revealed relative phase changes in water and/or ice content" -> "revealed relative changes in electrical resistivity related to phase changes in water and/or ice content"

Line 356-357: Did you measure the contact resistances? Did they change over time? And if yes, how strong? Did you have a look in the change of apparent resistivities over time?

Line 359: "Further, our modelled resistivities are in line with recent surface geophysical soundings near B3 and B4" Geophysical sounding means 1D survey. I guess you used a configuration for sounding and profiling, right? If yes, I would remove the term "sounding".

Line 361-362: The resistivities in Fig. 6 suggest that, even the temperature is <0°C below a depth of 4m, the material between 4 and 10 m depth is not frozen or there is only a little amount of ice. Could you try to give some possible explanations?

Line 364: "fines" -> "fine"

Line 365a: "ice from snow melt" -> " "ice from the time of the snow melt"

Line 365b: "presence of suprapermafrost water" The low resistivity anomaly is not only in the layer above 4 m depth. If you define the upper boundary of the permafrost layer by the temperature curve in Fig. 4, the water is not only suprapermafrost but also in the permafrost layer.

Line 366: "resistivities were highest through all seasons" Could you add the depth here please.

Line 367-69: Maybe you can refer here to the temperature profile in fig. 4. Maybe the water does not freeze until a depth of 10 m because the temperature is too high for freezing (-0.2°C) at the given salinity/Gibbs-Thomson effect? Or is there a preferential flow path?

Line 367: "permafrost resistivities" -> "resistivities"

Line 371: "temperature decrease and the" -> "temperature decrease in spring and the"

Line 373: What do you mean with striking here?

Line 383: "catalysor" -> "catalyst"

Line 393: "amount of unfrozen water" -> "amount of unfrozen water even in the permafrost layer"

Line 393-394: "The stratigraphic recordings and ERT data depict the small-scale heterogeneity within the rock glacier and the low resistivity anomalies throughout the ERT images." This sentence sounds confusing.

Line 399: "From 1991 to 2000, the depth of the shear zone in borehole B1 on rock glacier Ursina III was between 16.4 m and 11.4 m" -> "From 1991 to 2000, the shear zone in borehole B1 on rock glacier Ursina III was in a depth between 16.4 m and 11.4 m"

Line 414: "observe the relative water content changes" -> "observe the relative resistivity changes associated to water content changes"

Line 415-420: Could you try to make the paragraph a bit more clear?

Line 422: "permafrost temperatures below the active layer remained low." -> "temperatures in the permafrost layer remained low."

Line 428: What about vertical water flow?

---

## Referee Comment (RC2)

[referee-annotated manuscript omitted]

---

## Community Comment (CC1)

[revised manuscript text omitted]

---

## Author Comment (AC1)

Dear Professor Hördt

we appreciate your positive feedback regarding our manuscript titled "Short-term cooling, drying, and deceleration of an ice-rich rock glacier (egusphere-2024-269)" and thank you for your community comment. We will not provide a point-by-point response to your comments, but we will provide general responses below (our responses are denoted in blue). One of the two reviewers already raised some of your suggestions, and we have included them in our revised manuscript.

With kind regards,

Alex Bast, Marcia Phillips and Robert Kenner

Community Comment for Manuscript Number egusphere-2024-269

**Short-term cooling, drying and deceleration of an ice-rich rock glacier**

*Bast et al.*

This is a nice case study with a comprehensive characterisation of a rock glacier using a novel combination of methods. I find the results significant and useful. The material fits into the scope of the special issue and clearly deserves to be published.

Most of the results are clearly presented. In particular, I like detailed figure captions.

I have no issues with the analysis and interpretation of the data, but I believe the discussion and presentation of the results could be improved. Since this is a highly interdisciplinary subject, special efforts should be made to make sure that readers from all disciplines can follow. This includes explaining a little more than one normally would do, avoiding slang, and using precise wording and definitions. I have marked some sections that might be improved in this respect.

Thank you for your positive statement, your agreement with the interpretation and analysis of the data, and your assessment of the potential for publication. We followed most of your comments and included them in the text. This includes a clearer formulation of the abstract, the site description, and particularly the section on cross-borehole electrical resistivity tomography (chapter 2.4). We refer to the original publications for details, particularly within the methods section. Since we use a combination of mostly novel techniques for permafrost environments, we try to give enough detail to ensure the text is clear, detailed and short enough for a scientific publication. To our knowledge, the manuscript does not contain any slang.

I do not see the usefulness of figure 8, and suggest to remove it or replace it. In the current version, it is overloaded, and it is not clear which conclusions may be drawn from it that can not be made from other figures. For example, the resistivity images nicely show increase and decrease over the season, including spatial variations. By calculating an average over the entire volume, the information is being blurred. I suggest to re-think which message should be conveyed by the figure and redesign it correspondingly, or to remove it altogether.

Figure 8 summarizes the distribution and change of the variables ground temperature, piezometric pressure and resistivity over time and includes the robust statistical tests, which are highlighted and interpreted in the Discussion (L378-380) and Conclusions (L418-420). The figure and associated

statical tests show the importance of the late winter and spring months for rock glacier subsurface properties and, according to our interpretation, for rock glacier kinematics. Further, the plot shows the density distributions of the underlying sample and highlights the statistical key figures using traditional box plots, avoiding "blurring" and the pure presentation of classic descriptive figures such as arithmetic mean and standard deviation. We therefore consider this figure to be relevant and important. This is in line with Reviewer 2, who pointed out the added value of robust statistical data analysis.

I also have an issue with terminology; there seems to be confusion or imprecise usage of the term "active layer" and parameters related to it. The active layer is the layer below the surface that reaches temperatures above zero at least once during a season. It follows that the active layer thickness is the maximum depth of the thawed layer during a season. Therefore, the ALT cannot be measured at one point in time, and it cannot vary over a time scale of only one month or a few days. I recommend to be precise with terminology to avoid confusion for readers from other disciplines.

With regard to the term *active layer*, we refer to the uppermost part of the ground that thaws and refreezes on a seasonal basis. In our manuscript, we used the term *active layer thickness* (ALT), i.e. the maximum depth to which the 0°C isotherm penetrates in summer/autumn, where we explicitly wanted to refer to this state of the ground (we know the ALT from our presented data). However, for clarity, we have reformulated the caption of Fig. 5, stating that the white dashed line shows the *depth of thaw*.

I also recommend to consider this additional reference, a study with simular goals, but a slightly different combination of methods and a different region.

Buckel, J., Reinosch, E., Voigtländer, A., Dietze, M., Bücker, M., Krebs, N., Schroeckh, R., Mäusbacher, R., Hördt, A. 2022. Rock Glacier Characteristics Under Semiarid Climate Conditions in the Western Nyainqêntanglha Range, Tibetan Plateau. Journal of Geophysical Research: Earth Surface, 127, e2021JF006256. DOI: 10.5194/tc-15-149-2021.

---

## Author Comment (AC2)

Dear Professor Ntarlagiannis

we appreciate your positive feedback regarding our manuscript titled "Short-term cooling, drying, and deceleration of an ice-rich rock glacier (egusphere-2024-269)" and thank you for your review. We addressed your comments, outlined in the attached PDF document, and provide detailed responses to each point below (our responses are denoted in blue, with suggested changes highlighted in **bold**). We are confident that the modifications made to the manuscript, along with the explanations provided in our responses, address all your concerns and enhance the quality of our manuscript.

With kind regards,

Alex Bast, Marcia Phillips and Robert Kenner

Reviewer Comments for Manuscript Number egusphere-2024-269

**Short-term cooling, drying and deceleration of an ice-rich rock glacier**

*Bast et al.*

The manuscript discusses a multi year glacier monitoring project using a variety of monitoring tools. The manuscript is well written and logically organized. The authors provide a lot of information, collected data and robust statistical analysis, that could aid in describing the glacier kinematics.

Although not an expert in this field (cryosphere), data, processing and conclusions seem logical and supported by the data presented an analysis.

Thank you for your positive feedback on our study. Based on your statement, and to avoid any possible misunderstanding regarding the landform and the process area, we would like to underline the specificity of the landform, an ice-rich rock glacier. In contrast to glaciers, rock glaciers are landforms that are part of the periglacial process system, formed by the deposition of material from mass movements (snow avalanches and rock fall) at the base of steep slopes. Rock glaciers have a completely different internal structure than glaciers. Their structure typically includes layers of blocks, coarse-grained material, fines and ice, with varying contents of ice and/or water.

My only concern are the presented ERT data (my area of expertise). The environment is difficult to perform ERT with very high contact resistances (need to be described) that could degrade the quality of the data. The subsurface resistivity images provided, show an interesting, to say the least, structure that does not seem to change over time (very minor changes at certain time periods). The described subsurface stratigraphy does not fully explain/support such resistivity structure where the dominant conductive (relative) feature is oriented vertically - with significant lateral variability; the stratigraphy provided does not seem to support such variability in such close distance. The authors need to explain what subsurface stratigraphy can create such resistivity distribution, and they can achieve that through modeling. I fear that this resistivity image might be the result of systematic error with the ERT data acquisition, maybe due to poor contact resistance (or erroneous electrode mapping), that creates the observed anomaly and any variability detected could be the effect of water freezing/thawing on contact resistances.

I am not suggesting that the data are bad, but due to the local environmental conditions all the options

should be explored and discussed in the manuscript. Of course the glacier subsurface can be very variable that creates this resistivity image, but this need to be investigated.

We agree that it is difficult to carry out ERT measurements in such environments with very high resistances. This challenge can degrade the quality of ERT data, as the application of geophysical techniques is not as straightforward as their application in unfrozen, homogeneous and fine-grained soils. However, geophysical methods have been successfully applied in permafrost regions for decades (Kneisel et al., 2008; Scott et al.), and ERT is now one of the standard geophysical techniques for detecting and characterising permafrost (Hauck, 2013; Herring et al., 2023). Nevertheless, we are aware that geophysical surveys in these harsh environments require special techniques for sensor coupling, data acquisition and interpretation (Hauck and Kneisel, 2008).

To our knowledge, we are the first to apply cross-borehole ERT in permafrost and/or rock glacier environments. To improve electrode coupling, we used stainless steel electrodes integrated on a multi-core ERT cable in combination with steel clamps and used a finer-grained material (sand-gravel mixture) to establish contact between the electrodes and the borehole walls. We suggest that this be added to the Methods section, Chapter 2.4:

**"We used stainless steel electrodes (l = 100 mm; d = 13 mm) integrated on a multi-core ERT cable. To improve contact with the ground, we installed stainless steel clamp collars (h = 11 mm; D = 34 mm). To establish contact between the electrodes and the walls of the boreholes, we filled the boreholes with a mixture of sand (grain size ≤ 2 mm) and gravel (> 2 – ≤ 4 mm) in a ratio of 1:1."**

In terms of data collection, we are aware that ERT permafrost investigations are often susceptible to data quality problems associated with high contact resistance, and to obtain reliable inversions, it is important to remove the influence of poor-quality data (e.g., Hilbich et al., 2011). To do this, we apply a multi-stage filtering approach to our collected data. We believe that by collecting reciprocal data, we can improve data quality, as reciprocal data provides a good estimate of the precision of the resistivity data (Binley, 2015; Binley and Slater, 2020; Oldenborger and Leblanc, 2018).

Our ERT cross-borehole data show a relatively conductive feature that is vertically oriented with some variability. The reviewer, Dimitrios Ntarlagiannis, agrees that a rock "glacier subsurface can be very variable that creates this resistivity image, but this needs to be investigated". We believe that our manuscript clearly investigates and demonstrates that the ground in rock glaciers can indeed be very heterogeneous over small distances. We demonstrate this with the stratigraphic records of three boreholes drilled within a few metres of each other. In addition, at the same depth as the low resistivity structure, we detected a significant amount of unfrozen water (wet sludge layer) during drilling in 2020, which can cause the described low resistivities. We have added a new paragraph in the Discussion of our revised manuscript to emphasise the presence of unfrozen water content under frozen conditions. Furthermore, the ERT data are consistent with previous surface geophysical measurements by Boaga et al. (2020) and Pavoni et al. (2023) and with the piezometric pressure and temperature data presented in our manuscript. We attribute the small seasonal changes in resistivity within this wet sludge layer to latent heat effects and additionally refer to (Phillips et al., 2023). In general, the latter highlights the high potential of combining borehole temperature and piezometric pressure data with cross-borehole ERT data in rock-glacier environments.

We certainly agree that synthetic modelling can make a valuable contribution to better understanding cross-borehole ERT data and their interpretation for permafrost regions. However, this is outside the scope of our study, which presents measurements of relative water and/or ice content in rock glaciers for the first time, with the aim of understanding variations in rock glacier kinematics and the conditions leading to rock glacier deceleration. We are currently planning a more detailed study of cross-borehole ERT measurements in ice-rich permafrost, including different geometries, synthetic

modelling and sensitivity analyses, using data from this site and two others where we will be installing similar instrumentation this summer.

Minor comments on the annotated pdf:

L 45: truth

We have changed "ground truths" to "**ground truth**".

L110: Table 1 is missing

We have corrected the reference to "**Table S1**".

L139 & L146/147: What is an: ERT cross-borehole sounding? correct language

We now use "**cross-borehole ERT measurements**".

L154 – 160: ResIPy inverts for the difference of only common measurements - how can the background data have 355 data points, but thre is a subsequent inversion with more data points?

We ran the ResIPy routine according to the developer's instructions. The first model, the background data, was inverted independently using Occam's inverse method. Subsequent data sets subtracted the background data and inverted using the difference inversion algorithm. The data points remaining after applying our filters and reading in for the inversion routine are shown in parentheses.

To be clearer, we reformulated the text:

"ResIPy's time-lapse algorithm is based on the difference inversion method of Labrecque and Yang (2001). **Based on Occam's inverse method (Binley and Slater, 2020), the background data were inverted in the first step. Subsequent data sets subtracted the background data before inverting the data with the difference algorithm, which attempts to reduce the misfit between the difference in two data sets and the difference between two model responses (Labrecque and Yang, 2001)**. According to the reciprocal error check (Binley, 2015), fitting a power-law error model, and evaluating the normalized inversion errors (Blanchy et al., 2020), an expected data error of 10% was defined for the inversion process. **The 20 August 2021 model (ERT01) is used as a reference (background model), i.e. changes in resistivity are expressed as a percentage difference from this first reference survey. The inversion** converged after five iterations (final RMS misfit: 1.09; remaining data points: 355). All other models (ERT02 – ERT23) converged after a maximum of two iterations (average of remaining data points **after filtering**: 457; range: 242 – 635)."

L170/171: reference the relevant figures, or comment that they will be shown in the results section.

We included the reference to the relevant figures.

L252: all includes ERT01

We modified the text: "across all tomograms (**ERT01** to ERT23)".

Fig. 6: very resistive environment (as expected); what range were the contact resistances?

We agree that it would be useful to have the contact resistances, but unfortunately, we do not have information on contact resistances, which we stated in L356/357 and refer to Phillips et al. (2023).  As this was also raised by reviewer 1, we have already suggested to mention this challenge in the Discussion. We include our response to Reviewer 1 here:

*We discussed recording the contact resistance when setting up the experiment as it is an excellent idea. It is certainly possible to measure the contact resistance at each measurement. However, for energy reasons this is not feasible. We suggest to further discuss this topic in the Discussion section of the manuscript, and suggest adding the following paragraph:*

*"Despite uncertainties in the contact between electrodes and ground material (Phillips et al., 2023), the consistent structure in all tomograms and the similarity to the August 2020 borehole stratigraphies (Phillips et al. 2023) across the summer images promotes confidence in data reliability. **To avoid the uncertainty of the contact resistance between the electrode and the substrate material, it is in principle possible to record the contact resistances for each ERT measurement. However, we run the system on a solar panel in a harsh alpine environment, with sufficient power for a daily measurement including data transfer, which already reaches the limits of the power supply during the winter season, i.e. data transfer is not always possible. To avoid data loss due to power supply, we have accepted not to record the contact resistance. However, we do record reciprocal data to gain information on data quality (see chapter 2.4), and** our modelled resistivities are in line with recent surface geophysical measurements near B3 and B4 (Boaga et al., 2020; Pavoni et al., 2023)."*

Fig. 6: I would suggest removing log10 from the colro bar, and change the number to actual resistvity (e.g. 1000, 10000 etc) makes the graph more readable.

For better readability we restyled the colour bar and added the actual resistivities next to the log10 numbers on kΩm.

L383: catalyst?

We corrected the word as suggested.

L392: not that evident from the ERT data

We refer to our general response above and believe that our ERT data, presented in combination with ground temperature and piezometric pressure data, confirm that rock glaciers at or near their melting point can contain a considerable amount of unfrozen water.

References

Binley, A.: 11.08 - Tools and Techniques: Electrical Methods, in: Treatise on Geophysics (Second Edition), edited by: Schubert, G., Elsevier, Oxford, 233-259, https://doi.org/10.1016/B978-0-444-53802-4.00192-5, 2015.
Binley, A. and Slater, L.: Resistivity and Induced Polarization: Theory and Applications to the Near-Surface Earth, Cambridge University Press, Cambridge, DOI: 10.1017/9781108685955, 2020.
Boaga, J., Phillips, M., Noetzli, J., Haberkorn, A., Kenner, R., and Bast, A.: A Comparison of Frequency Domain Electro-Magnetometry, Electrical Resistivity Tomography and Borehole Temperatures to Assess the Presence of Ice in a Rock Glacier, Frontiers in Earth Science, 8, 10.3389/feart.2020.586430, 2020.
Hauck, C.: New Concepts in Geophysical Surveying and Data Interpretation for Permafrost Terrain, Permafrost Periglac, 24, 131-137, https://doi.org/10.1002/ppp.1774, 2013.
Hauck, C. and Kneisel, C., Hauck, C., and Kneisel, C. (Eds.): Applied Geophysics in Periglacial Environments, Cambridge University Press, Cambridge, DOI: 10.1017/CBO9780511535628, 2008.
Herring, T., Lewkowicz, A. G., Hauck, C., Hilbich, C., Mollaret, C., Oldenborger, G. A., Uhlemann, S., Farzamian, M., Calmels, F., and Scandroglio, R.: Best practices for using electrical resistivity tomography to investigate permafrost, Permafrost Periglac, 34, 494-512, https://doi.org/10.1002/ppp.2207, 2023.

Hilbich, C., Fuss, C., and Hauck, C.: Automated Time-lapse ERT for Improved Process Analysis and Monitoring of Frozen Ground, Permafrost Periglac, 22, 306-319, https://doi.org/10.1002/ppp.732, 2011.

Kneisel, C., Hauck, C., Fortier, R., and Moorman, B.: Advances in geophysical methods for permafrost investigations, Permafrost Periglac, 19, 157-178, 10.1002/ppp.616, 2008.

Oldenborger, G. A. and LeBlanc, A. M.: Monitoring changes in unfrozen water content with electrical resistivity surveys in cold continuous permafrost, Geophys J Int, 215, 965-977, 10.1093/gji/ggy321, 2018.

Pavoni, M., Boaga, J., Wagner, F. M., Bast, A., and Phillips, M.: Characterization of rock glaciers environments combining structurally-coupled and petrophysically-coupled joint inversions of electrical resistivity and seismic refraction datasets, Journal of Applied Geophysics, 215, 105097, https://doi.org/10.1016/j.jappgeo.2023.105097, 2023.

Phillips, M., Buchli, C., Weber, S., Boaga, J., Pavoni, M., and Bast, A.: Brief communication: Combining borehole temperature, borehole piezometer and cross-borehole electrical resistivity tomography measurements to investigate seasonal changes in ice-rich mountain permafrost, The Cryosphere, 17, 753-760, 10.5194/tc-17-753-2023, 2023.

Scott, W. J., Sellmann, P. V., and Hunter, J. A.: 13. Geophysics in the Study of Permafrost, in: Geotechnical and Environmental Geophysics: Volume I, Review and Tutorial, 355-384, 10.1190/1.9781560802785.ch13,

---

## Author Comment (AC3)

Dear Referee 1

we are very grateful for your positive, constructive and detailed feedback on our manuscript *Short-term cooling, drying and deceleration of an ice-rich rock glacier* (egusphere-2024-269). - We have carefully considered all your comments and provide our point-by-point responses below (our responses are in blue; suggested changes in **bold**). We believe that the changes to the manuscript and the justifications provided in our responses fully address all your concerns and improve the quality and readability of our manuscript.

Sincerely,

Alex Bast, Marcia Phillips and Robert Kenner

Reviewer Comments for Manuscript Number egusphere-2024-269

**Short-term cooling, drying and deceleration of an ice-rich rock glacier**

*Bast et al.*

**Summary**

This highly relevant and interesting manuscript presents the combination of borehole temperature, piezometric pressure and geophysical data with surface meteorological and kinematic data to investigate a short-term deceleration of a rock glacier in the Swiss Alps. To the best of my knowledge, the use of borehole ERT combined with borehole temperature and piezometric pressure data in a rock glacier is a novelty and makes the manuscript very interesting.

The results of the study show that the snow cover plays a crucial role for short-term velocity changes due to its effects on the thermal state and particularly due to the water supply. According to this manuscript, snow-poor winters and dry summers are ideal for the short-term deceleration of the rock glacier under investigation. Additionally, the authors demonstrated that a summer heat wave did not play a significant role in the rock glacier movement.

The manuscript is well-written, well-structured and the figures are well designed. The Introduction provides a good overview about the relevance of the study, what has been done so far and the main objectives of the manuscript. In Material and Methods, the authors provided all information needed to understand the data collection, the processing, statistical analyses and visualizations, but a few points in the description of the ERT data processing are missing or should be revised. The results and the discussion are well presented. However, in the discussion the authors could try to focus on explaining links and the (somehow surprising) discrepancy between temperature and resistivity in the permafrost layer. These two issues together with some specific comments and technical corrections (see below) can be easily addressed. Therefore, I recommend accepting this interesting manuscript with minor revisions.

Thank you again for your very positive and constructive feedback. Based on your suggestions below, we believe we have addressed your two main concerns regarding the description of ERT data processing and suggestions for improving the discussion.

**General Comments**

Processing of electrical resistivity data

- See specific comments below

Discussion of the low resistivity zone in the permafrost layer

- See specific comments below

**Specific comments and technical corrections**

Line 9: „Long term" -> „Long-term" (for consistency)

For consistency we have changed "long term" to "**long-term**" throughout the paper.

Line 10: "variability of their velocity and there is still a gap in the" -> "variability of the velocity with a particular gap in the"

We have changed the sentence as suggested.

Line 11: "role of water in rock glaciers" Role in what? Maybe add here "in rock glacier movement"

We agree that the "role in what" was missing and we have added "the role of water in rock glacier **kinematics**" to the sentence.

Line 14a: "cross-borehole electrical resistivity tomography sensors." -> "cross-borehole electrodes for electrical resistivity tomography measurements"

We agree and suggest to change the sentence: "cross-borehole **electrodes** for electrical resistivity tomography **measurements**".

Line 14b: "using terrestrial" -> "using repeated terrestrial"

We have added "using **repeated** terrestrial" for clarity.

Line 17: "lowering of the water content in rock glaciers is crucial for" Where exactly? In general, at the shear horizon, at the surface?

We agree and have changed the sentence to "lowering of the water content in rock glacier **shear horizons** is crucial".

Line 22: "ice/water contents" -> "ice/water content"

We have changed the sentence to "ice **and/or** water contents", leaving the plural.

Line 23: "better understanding of factors affecting rock glacier kinematics" -> "better understanding of the main drivers of rock glacier kinematics"

We agree and have changed the sentence as suggested.

Line 26: See also the recently published paper: Kellerer-Pirklbauer, A., Bodin, X., Delaloye, R., Lambiel, C., Gärtner-Roer, I., Bonnefoy-Demongeot, M., Carturan, L., Damm, B., Eulenstein, J., Fischer, A., Hartl, L., Ikeda, A., Kaufmann, V., Krainer, K., Matsuoka, N., Morra Di Cella, U., Noetzli, J., Seppi, R., Scapozza, C., Schoeneich, P., Stocker-Waldhuber, M., Thibert, E., and Zumiani, M.: Acceleration and interannual variability of creep rates in mountain permafrost landforms (rock glacier

velocities) in the European Alps in 1995–2022, Environ. Res. Lett., 19, 034022, https://doi.org/10.1088/1748-9326/ad25a4, 2024.

Thank you for pointing out this relevant reference, which we have now included.

Line 27: "Rock glacier kinematics are driven by common external climatic factors, and they vary over time (Delaloye et al., 2010a; Delaloye et al., 2008), with phases of acceleration, interrupted by periods of stagnation or deceleration of variable duration (Permos, 2023a)." -> "Rock glacier kinematics vary over time (Delaloye et al., 2010a; Delaloye et al., 2008), with phases of acceleration, interrupted by periods of stagnation or deceleration of variable duration (Permos, 2023a), driven by common external climatic factors.

As our intention was to point out the drivers first, we suggest leaving the sentence as it was.

Line 37-40: At this position, the paragraph is a bit lost and disconnected. Can you add it in the first paragraph after the first sentence?

We agree and have moved the paragraph after the first sentence of the introduction.

Line 41: Can you specify the type of space-borne data?

We have changed the sentence to "Analyses of rock glacier kinematics are often carried out using remote sensing data, which, **for instance in the case of spaceborne interferometric synthetic aperture radar (InSAR),** even allow quantification of the rate and direction of rock glacier creep at a global scale". Further, we changed the reference to **Bertone et al. 2022** since we inadvertently referred to the publication from Bertone et al. 2023:

Bertone, A., Barboux, C., Bodin, X., Bolch, T., Brardinoni, F., Caduff, R., Christiansen, H. H., Darrow, M. M., Delaloye, R., Etzelmüller, B., Humlum, O., Lambiel, C., Lilleøren, K. S., Mair, V., Pellegrinon, G., Rouyet, L., Ruiz, L., and Strozzi, T.: Incorporating InSAR kinematics into rock glacier inventories: insights from 11 regions worldwide, The Cryosphere, 16, 2769-2792, 10.5194/tc-16-2769-2022, 2022.

Line 47: "rock glacier, and recent literature" -> "rock glacier. Recent literature"

We agree and have split the sentence into two sentences for better readability.

Line 53: "refraction seismic" -> "refraction seismic tomography (SRT)"

We have modified the wording as suggested.

Line 55: In the references, maybe you can add: Kneisel, C., Hauck, C., Fortier, R., and Moorman, B.: Advances in geophysical methods for permafrost investigations, Permafrost Periglac., 19, 157–178, https://doi.org/10.1002/ppp.616, 2008.

We have added the reference to Kneisel et al. 2008 (see our response to Lines 52 – 62 below).

Line 52-62: Could you try to restructure the paragraph? Maybe you start with the direct method (borehole temperature sensors), their limitations and present the geophysical methods with their advantages over the direct methods as alternatives? What about the spatial coverage/resolution? That would make it clearer why you use a geophysical method in your study.

Based on your suggestions we will restructure and modify the paragraph thus:

"When moving from rock glacier kinematics to their internal characteristics (Arenson et al. 2002), **direct subsurface data from rock glaciers is quite scarce, for logistic and financial reasons. Borehole temperatures are currently the most widespread type of subsurface data, allowing to monitor the temperature regime, active layer thickness/duration and thermal anomalies, such as those**

triggered by water or air fluxes (Noetzli et al., 2021; Permos, 2023b; Luethi and Phillips, 2016; Zenklusen Mutter and Phillips, 2012b). Nevertheless, temperature data alone do not allow to discern between ice and water, which can coexist at 0 °C and information on water content is essential to understand rock glacier kinematics. Applied near-surface geophysics such as electrical resistivity tomography (ERT), refraction seismic tomography (SRT), ground penetrating radar (GPR) and electromagnetic methods deliver valuable information on rock glacier internal structure and the distribution of rock, air, ice and water (Hauck, 2013; Boaga et al., 2020; Pavoni et al., 2023; Hauck et al., 2011; Kneisel et al., 2008). Cross-borehole geophysics provide higher-resolution information on the near-surface structure (Binley and Slater, 2020) and are a means to distinguish ice and water in ice-rich permafrost and to monitor their volumetric variation and distribution over time by, e.g., electromagnetic velocity structures (GPR; traveltime tomography; Musil et al., 2006) or detecting changes with inverted resistivity models (Musil et al., 2006; Phillips et al., 2023)."**

Line 60-62: "Cross-borehole geophysics are a means to distinguish ice and water in ice-rich permafrost and to monitor their volumetric variation and distribution over time (Musil et al., 2006; Phillips et al., 2023)." Geophysical methods do not provide direct information about ice and water content, you still need a model linking a geophysical variable, like resistivity, to ice or water content. Please add that here.

We have provided more detailed information including a new reference, Binley and Slater 2020 (see our response / bold text above).

Line 73: I would remove the apostrophe in the elevation numbers.

We have removed the apostrophe from the numbers.

Line 77-81: The description of the boreholes and their location is a bit confusing. For example: Are there also boreholes B1 and B2? What is the depth of the boreholes? Did you reach the bedrock in boreholes B4 and B5 as well or just in B3? What  Please try to improve the structure.

We understand the initial confusion caused by the numbering of the boreholes. B1 is the PERMOS borehole with the thermistor chain (we added the borehole names); B2 is also a PERMOS borehole, but on the Ursina II rock glacier – and not mentioned in this paper. To be consistent with PERMOS and existing publications, we would like to keep the names of the boreholes (without changing their numbering for this publication). For more clarity on the location and stratigraphy of the boreholes drilled in 2020, we have created the two figures 1 (especially Fig. 1e) and 2.

Regarding the question of whether we reached the bedrock we will add more information to the text:

"Destructive drillings in 1990 **(B1)** (Vonder Mühll and Holub, 1992) and 2020 **(B3 to B5)** (Phillips et al., 2023) at Ursina III rock glacier **(Fig. 1e)** show that the uppermost 3 – 4 m consist of large boulders (Fig. 2), below which icy sediments and dirty ice are found (B3, Fig. 2). In two boreholes drilled in 2020 (B4 & B5) wet sludge with ice dominates below the blocky layer. Ice and water distribution are heterogeneous over distances of 5 to 10 m (Phillips et al., 2023). In 1990, bedrock was reached at a depth of 16 m (Vonder Mühll and Holub, 1992). **In 2020, bedrock was not reached (Fig.2)."**

Line 83: "relative ice-/water contents" The ERT method does not provide ice-/water contents directly, just a physical parameter.

We have changed the sentence to "pore water pressure (borehole piezometers), **ground resistivities** (cross-borehole ERT)".

Fig. 1: I would recommend to merge c and d, remove e, and add the scale bars and symbologies in the accompanying subplot. Additionally, the borehole names B2-B4 are missing. And I guess the north arrow is not valid for all subplots, right?

From our point of view, Figure 1c provides a wider perspective of the study area including the steep rock faces/cirque and surrounding mountain peaks, whereas the ortho image provides a more detailed and georeferenced subplot including the boreholes and GST locations. We therefore prefer to keep the two subplots separate. We also intend to retain the borehole location map (Figure 1e), as it could be very valuable in providing more detail on the design and location of the study site. For example, the small-scale heterogeneity becomes clearer once again. However, we included the borehole names as suggested. Borehole B2 is part of the PERMOS network but was not used in our study (see our comment above). The north arrow applies to all subplots. We added this information to the figure captions **("The north arrow is valid for all subplots.")**.

Fig. 2: "Syscal system" In the caption of the Figure, the type of electrodes would be more relevant than the instrument name.

We have modified the figure captions for clarity.

Line 110: Please add that the Table is in the Supplementary Material.

We have modifed the reference to the table to "Table **S**1".

Line 118: "These delivered" -> "They deliver"

We have changed the beginning of the sentence to "**They delivered**".

Line 122a: "icing days" -> "number of icing days"

We have modified the wording to: "as well as **number of** frost days, icing days,…".

Line 122b: "15 °C days" -> "15 °C-days

We have added the hyphen: "15 °C-days".

Line 126-127 and 130: Please add the information about the boreholes in section 2.1. What is in B1 below 15.2 m? Is it broken?

For clarity, we have rewritten the sentence: "**In 2005, after the original thermistor chains were sheared off, the borehole was instrumented with** 16 YSI thermistors of type 44006 with an accuracy of ± 0.1 °C (YSI Inc., Yellow Springs, OH, USA, www.ysi.com) to a depth of 15.2 m (Fig. 2)."

Line 134-136: Why B1 for the annual variations and B5 for the short-term variations?

We do not have a longer data series for B5 (since we drilled it in 2020). However, we wanted to highlight the peculiarities (highest temperatures in the active layer in 2022, but lower temperatures in the underlying permafrost than in 2020 and 2021; highest MAGT in 2021) of the last decade, so we had to show the data from the nearby PERMOS borehole B1 for the annual temperature variations. One of our objectives was to compare and interpret the piezometer sensors to investigate the importance of the water content on rock glacier movement. The piezometer sensors have a PT1000 temperature sensor built in. This allowed us to directly compare piezometric pressure and temperature in the same borehole. Hence, we used the data from B5 to illustrate the short-term variations.

Line 135: "the evolution" -> "short-term evolution"

We have added "**short-term** evolution".

Line 138: The abbreviation ERT has been already explained.

We have now only used the abbreviation.

Line 139: "ERT soundings" -> "ERT measurements"

We have renamed "soundings" to "**measurements**".

Line 140: "in total 23 ERT measurements" -> "in total 23 time steps"

We have changed the sentence as suggested: "in total **23 time steps**".

Line 146a: Please remove the apostrophe in "1494"

We have removed the apostrophes.

Line 146b: "data points" -> "data"

We have kept "data points" since we obtained 1494 individual data points.

Line 146c: "ERT sounding" -> "time step"

We have removed the word sounding and replaced it with: "for each ERT **time step**".

Line 147: Please describe what dipole-dipole and skip-two means.

The skip represents the number of electrodes skipped to create a dipole. In our case we used a dipole-dipole skip-two configuration, which means a dipole spacing of three electrodes (skip 0 would mean that no electrode was skipped, i.e. a spacing of one electrode). For clarification we have reformulated the sentence: "1494 direct and reciprocal data points were collected for each ERT **time step** with a dipole-dipole skip-two configuration **(dipole spacing of three electrodes)**."

Line 138-147: please add some information about contact resistances and their change over the whole year. I guess they are not constant over the time and might affect the data quality, particularly during winter months. Which type of electrodes did you use and how are they connected to the subsurface material?

Unfortunately, we do not have information on contact resistance. We stated this in L356/357 and refer to Phillips et al. (2023). We discussed to record the contact resistance when setting up the experiment as it is an excellent idea. It is certainly possible to measure the contact resistance at each measurement. However, for energy reasons this is not feasible. We suggest to further discuss this topic in the Discussion section of the manuscript, and have added the following paragraph:

"Despite uncertainties in the contact between electrodes and ground material (Phillips et al., 2023), the consistent structure in all tomograms and the similarity to the August 2020 borehole stratigraphies (Phillips et al. 2023) across the summer images promotes confidence in data reliability. **To avoid the uncertainty of the contact resistance between the electrode and the substrate material, it is possible to record the contact resistances for each ERT measurement. However, we run the system on a solar panel, with just enough power for a daily measurement and for data transfer. This already reaches the limits of the power supply during the winter season, i.e. data transfer is not always possible. To avoid data loss due to scant power, we have not recorded the contact resistance. However, we do record reciprocal data to gain information on data quality (see chapter 2.4), and** our modelled resistivities are in line with recent surface geophysical measurements near B3 and B4 (Boaga et al., 2020; Pavoni et al., 2023)."

Regarding the types of electrodes and their connection to the ground material, we have added more details in the Methods section:

"**We used stainless steel electrodes (l = 100 mm; d = 13 mm) integrated on a multi-core ERT cable. To improve contact with the ground, we installed a stainless steel clamp collar (h = 11 mm; D = 34 mm). To establish contact between the electrodes and the walls of the boreholes, we filled the boreholes with a mixture of sand (grain size ≤ 2 mm) and gravel (> 2 – ≤ 4 mm) in a ratio of 1:1.**"

Line 149a: Please explain what reciprocal counterpart means.

We have rewritten the sentence: "Only (i) paired data, **i.e. data with a direct and a reciprocal measurement**".

Line 149b: "positive values of measured apparent electrical resistivities" -> "positive apparent electrical resistivities"

We have deleted "values of measured": "**(ii) positive apparent electrical resistivities**".

Line 148-150: How many quadrupoles did you keep for the inversion? Did you filter for every time step the same quadrupoles? If not, the combination of different quadrupoles might result in different sensitivities increasing the generation of time-lapse artifacts.

We tried to retain as many data points as possible based on the filter criteria described (L149 - 150). We used this filter for each ERT time step, resulting in an average of 457 data points (L158). In this study, in which we used a novel combination of methods for permafrost environments, we did not investigate the possible effects of combining different quadrupoles on the generation of time-lapse artefacts. This was beyond the scope of our investigations. However, we agree that this is a very interesting question and are currently planning a more detailed study of cross borehole ERT in permafrost environments, including different geometries, synthetic modelling and sensitivity analyses.

Line 150-152: 10 m mesh extension is a bit too small and might influence the inversion.

The distance between the two boreholes is 5 metres. Similar to Phillips et al. (2023) and Blanchy et al. (2020), the mesh size was doubled laterally and downwards. However, it was not part of our study and the aim of this paper to test different mesh types and sizes for cross-borehole ERT application on rock glaciers. We have included Phillips et al. (2023) as a reference.

Line 152: "ResIPy's" For inversions, ResIPy calls the R2 code – maybe you can add that here.

We have added the information about the R2 code to the manuscript after the first sentence in the second paragraph in the chapter 2.4: "**ResIPy uses the mature R2 code to invert ERT data.**"

Line 154: What means "according to the reciprocal check"? Could you specify the error model here? Is your error value (10%) similar to the one estimated from the normal-reciprocal analysis and is the error constant over all time steps? And why is your used error parameter (10%) much higher than your filtering threshold (25%)?

We used the reciprocity of as a measure of data quality. This involves switching the transmitter and receiver pairs. Before and after the switch, the pairs should be identical. The difference between them is called the reciprocal error. Therefore, the term "reciprocal error check" refers to filtered data with a reciprocal error of less than 25% for the inversion (see L150). Values of this magnitude are also used in other permafrost studies. ERT measurements in permafrost are very demanding (coupling, very high resistances, etc.). For this reason, the 25% limit was chosen as opposed to the highly conductive subsurface. On this basis, we fitted a power-law error model and kept the error constant over all

inversion time steps (as suggested by, e.g., Binley et al. 2020). We then analysed the normalised inversion errors which should range between 3% (Binley. Based on this, we successively improved our expected error for the inversion process leading to an expected data error of 10%.

Line 156a: Could you explain what is the reference model in this context?

We will be clearer and suggest changing the sentence to: "**The 20 August 2021 model (ERT01) is used as a reference, i.e. changes in resistivity are expressed as a percentage difference from this first reference survey.**"

Line 156b: "It converged" -> "The inversion converged"

We have changed the sentence as suggested.

Line 157: Did you use an error-weighted RMS? If yes, you could add that here.

As common for the R2 code, we expressed data misfit as a root mean square error. The final RMS misfit is given in the text (L157).

Line 168: "systemic" -> "systematic"

We have corrected the typo.

Line 171-179: Could you shift the paragraph with the description of the Figure to the results section?

From our point of view, this paragraph describes a (visualisation) method we used (violin plots), including the relevant data. For this reason, we prefer to leave the generic description of the violin plots used in the methods section.

Line 176: "entire layer" What do you mean with entire layer?

We agree and have reworded the sentence to be more precise: "Resistivities were extracted from the inverted resistivity models **between 2.0 – 8.5 m**".

Line 180: "differences of the variables" Which variables?

We have reformulated the sentence: "The robust non-parametric Kruskal-Wallis H-test statistic was applied to determine year-to-year **differences of the variables ground temperature, piezometric pressure, and electrical resistivity per month**."

Fig. 3: The temperature and precipitation data look very similar between the different stations. Could you try to present every parameter only for one station and shift the data of the other stations to the Supplementary material? That would make the statement of the Figure more concise.

We agree. The weather data shows similar patterns over all stations. We have therefore prepared two new/modified figures. One for the supplementary material (BER2 station; Figure S1) and one for the manuscript (all other stations and GST data; Figure 3). We have kept the IMIS station BER3 in the manuscript figure, as it is closer to the Ursina rock glacier. We also show the data from the Schafberg weather station (WS) as it is the only weather station on site. We have modified the captions and references in the text accordingly.

Line 211: "(ERT) soundings" -> "(ERT) measurements"

We have reformulated "soundings" into "**measurements**".

Line 241: "thickness of the active layer (ALT)" -> ""depth of the active layer"

We are referring to the maximum depth to which the 0°C isotherm penetrates in summer/autumn. This is defined as the thickness of the active layer (ALT). Hence, we suggest keeping ALT.

Line 248: "from the surface to ~ 10 m depth" -> "from the surface on the left hand side to ~ 10 m depth at the right hand side"

We have modified the sentence as suggested: "ERT01 revealed a distinctive zone with low resistivities from **the surface of the left hand side to ~ 10 m depth on the right hand side** (< 13 kΩm; orange-brown colors)."

Line 249a: "low and high resistivity segment" Please add the range of the values.

We have added the range of values: "The active layer had a low **(~ 1 kΩm; top left)** and high **(~ < 100 kΩm, bottom right)** resistivity segment.".

Line 249b: "Below the AL" Please add here the depth of the AL.

We have added the depth information: "High resistivity areas appeared below the **AL at ~ 4 m**".

Line 249c: "existed" -> "exists"

As we have written the entire results section in the past, we suggest leaving the word "existed" here.

Fig. 6: What about the sensitivity in the different time steps and in your model domain? To be more concise, you could merge Fig. 6 with Fig. 7 and keep the absolute values only for the reference time step or one image per season. In the colorbar please write Ωm instead of ohm.m.

We think the readability is much better in two separate figures (otherwise it would become a very large figure - similar to Figure 3, which we have now split). Further, to remain consistent, we would like to show all 23 ERT time steps in Figure 6. Hence, we would prefer to keep the two figures as they are. We have replaced ohm.m with Ωm.

We did not perform a detailed sensitivity analysis for the different time steps for this study. However, we are planning a more detailed study on cross-borehole ERT in permafrost environments, including different geometries, synthetic modelling, and sensitivity analysis.

Line 256: "cross-borehole electrical resistivity tomography (ERT)" -> "cross-borehole ERT"

We have used the abbreviation as suggested.

Line 259-260: "The ERT multi-core cables with 24 electrodes per borehole (indicated by the light grey short lines) and a vertical spacing of 0.5m are installed the boreholes B3 and B4." You wrote that information already in the Methods. You can change that to: "Electrode positions are indicated by light grey short lines.

We have removed the repetition and changed the figure captions of figure 6 and figure 7 as suggested.

Line 262: "See Fig. 1 for borehole locations, Fig. 2 for further details and stratigraphy, and Fig. 7 for relative electrical resistivity changes." Not necessary here.

We have removed the cross-references.

Line 277-278: "See Fig. 1 for borehole locations, Fig. 2 for further details and stratigraphy, and Fig. 7 for inverted resistivity tomograms." Not necessary here.

We have removed the cross-references.

Fig. 8: "kohm.m" -> "kΩm"

We have used the suggested notation.

Line 04: "fastest" -> "highest"

We agree and have changed the wording as suggested.

Line 309a: "TLS were collected" -> "TLS data were collected"

We have changed the wording as suggested.

Line 309b: "each year" -> "each year from 2018 to 2023"

We have added the time period: "TLS **data** were collected in July of each **year from 2018 to 2023.**".

Line 328: "Changes in temperature T" -> "Changes in air temperature TA"

We have modified the wording as suggested.

Fig. 10: In these Figure you use different colors for two different categories which makes it confusing. It would be easier to understand the plot, if you directly write down which column presents winter and which one summer and you keep the bars in the same color.

We have plotted the summer and winter bars in the same colour. As suggested, we have labelled summer and winter directly on the bars. We have also changed the colour of the drier years to blue, thus keeping the same colour for precipitation. We have modified the captions accordingly.

Line 343: I think "SZC" has not been introduced before.

We have added "**spring zero curtain** (SZC)" to the sentence.

Line 355a: "electrical resistivity tomography (ERT)" -> "ERT"

We have used the abbreviation as suggested.

Line 355b: "revealed relative phase changes in water and/or ice content" -> "revealed relative changes in electrical resistivity related to phase changes in water and/or ice content"

We have changed the sentence to: "The monthly cross-borehole **ERT data revealed relative changes in electrical resistivity related to phase changes in water and/or ice content** across a 5 x 11.5 m area, complementing the piezometer data".

Line 356-357: Did you measure the contact resistances? Did they change over time? And if yes, how strong? Did you have a look in the change of apparent resistivities over time?

We refer to our comment above (Line 138-147) and suggest adding a new paragraph in the Discussion section of the manuscript.

For this manuscript we have not considered the change in apparent resistivity with time. We plan to analyse the change in apparent resistivity over time in a follow-up paper that will include cross-borehole ERT measurements (on a daily basis) as well as piezometer and temperature data, focusing on snowmelt and rainfall events.

Line 359: "Further, our modelled resistivities are in line with recent surface geophysical soundings near B3 and B4" Geophysical sounding means 1D survey. I guess you used a configuration for sounding and profiling, right? If yes, I would remove the term "sounding".

We have changed the term "sounding" to "recent surface geophysical **measurements** near".

Line 361-362: The resistivities in Fig. 6 suggest that, even the temperature is <0°C below a depth of 4m, the material between 4 and 10 m depth is not frozen or there is only a little amount of ice. Could you try to give some possible explanations?

We mention the indication of a considerable amount of water and point out the presence of sludge containing ice crystals. Despite the ground being at subzero temperatures, it contains a significant amount of unfrozen water. We include this in the discussion and provide a much more detailed response below in response to your comment on lines 367-369.

Line 364: "fines" -> "fine"

We suggest to leave "fines" in the manuscript since "fines" is the term used for fine-grained ground material.

Line 365a: "ice from snow melt" -> " "ice from the time of the snow melt"

We agree and will change the wording as suggested.

Line 365b: "presence of suprapermafrost water" The low resistivity anomaly is not only in the layer above 4 m depth. If you define the upper boundary of the permafrost layer by the temperature curve in Fig. 4, the water is not only suprapermafrost but also in the permafrost layer.

We agree with this comment and will refer to it in the Discussion section (Lines 366 – 370). However, we will also include more details in the Discussion. We provide a much more detailed response below in response to your comment on lines 367-369.

Line 366: "resistivities were highest through all seasons" Could you add the depth here please.

We have added the depth here: ". In deeper ice-bearing layers **(~ 4 to ~ 6 m, and from ~ 9 down)**, resistivities were highest through all seasons."

Line 367-69: Maybe you can refer here to the temperature profile in fig. 4. Maybe the water does not freeze until a depth of 10 m because the temperature is too high for freezing (-0.2°C) at the given salinity/Gibbs-Thomson effect? Or is there a preferential flow path?

We agree and have referred to Figures 4 and 5a.

We also agree on the presence of unfrozen water below 0°C, either due to salinity and/or to pressure. In the discussion, we include the effect of temperature (due to differences in snow cover) and conclude that "Rock glacier deceleration appears to be primarily caused by ground cooling and drying, resulting from winters with little snow. These conditions lead to a decrease in ground water content, as the ground can freeze efficiently in the absence of an insulating snow cover." (L415-417).

However, we suggest that the following be added to the Discussion section: "**The presence of unfrozen water , i.e. liquid water in frozen ground at temperatures below the phase equilibrium temperature (Romanovsky and Osterkamp, 2000), due to pressure conditions, grain size, saline water and / or other soil properties, has been quantified in several laboratory (e.g., Williams, 1964), field experiments (e.g., Oldenborger and Leblanc, 2018) and in modelling studies (e.g., Bi et al., 2023). However, less is known about the amount of unfrozen water in coarse-grained ground, particularly in rock glaciers, and our data suggest intrapermafrost water fluxes.**".

With the methods and temporal resolution used in this manuscript, we did not identify clear preferential flow paths of water. However, based on our data, we are convinced that there is intrapermafrost water (flow). The detection of preferential flow paths was beyond the scope of the

present study. We will continue to work on this in a planned follow-up paper with detailed analyses of ground water in relation to snow melt and heavy rainfall events.

Line 367: "permafrost resistivities" -> "resistivities"

We agree and will delete "permafrost".

Line 371: "temperature decrease and the" -> "temperature decrease in spring and the"

We have added: "with the subsequent temperature decrease **between January and June** and the".

Line 373: What do you mean with striking here?

We will change the wording and replace the word "striking": "The relative resistivity changes in the tomograms below ~ 9 m were **prominent**."

Line 383: "catalysor" -> "catalyst"

We have modified the wording as suggested.

Line 393: "amount of unfrozen water" -> "amount of unfrozen water even in the permafrost layer"

We partially agree and have changed the sentence to: "The piezometric pressure and ERT data confirm that rock glaciers at or near their melting point can contain a substantial amount of **unfrozen water in the permafrost.**"

Line 393-394: "The stratigraphic recordings and ERT data depict the small-scale heterogeneity within the rock glacier and the low resistivity anomalies throughout the ERT images." This sentence sounds confusing.

We suggest the following: "The stratigraphic recordings and ERT data **represent** the small-scale heterogeneity within the rock glacier, **i.e. the low resistivity anomalies throughout the ERT images indicating a considerable amount of unfrozen water at around 7 m depth.** This suggests there is the potential for water to flow from the active layer into deeper layers via preferential pathways, **that is intrapermafrost water flow.**"

Line 399: "From 1991 to 2000, the depth of the shear zone in borehole B1 on rock glacier Ursina III was between 16.4 m and 11.4 m" -> "From 1991 to 2000, the shear zone in borehole B1 on rock glacier Ursina III was in a depth between 16.4 m and 11.4 m"

We have changed the sentence as suggested: "From 1991 to 2000, **the shear zone in borehole B1 on rock glacier Ursina III was at a depth between 16.4 m and 11.4 m,**"

Line 414: "observe the relative water content changes" -> "observe the relative resistivity changes associated to water content changes"

We have changed the sentence to: "Our monitoring approach allowed us to **observe the relative resistivity changes associated to water content changes** of rock glaciers".

Line 415-420: Could you try to make the paragraph a bit more clear?

We rewrote the paragraph for clarity: "**Rock glacier deceleration appears to be primarily caused by ground cooling and drying. Winters with little snow lead to decreases in ground water content, as the ground can freeze efficiently in the absence of an insulating snow cover. Low water contents in late winter and early spring are the main driving factor for rock glacier deceleration. Differences in piezometric pressure, ground temperature and ground resistivities were only statistically significant at these times of year.**"

Line 422: "permafrost temperatures below the active layer remained low." -> "temperatures in the permafrost layer remained low."

We have changed the sentence as suggested: "and **temperatures in the permafrost remained low.".**

Line 428: What about vertical water flow?

We have not been able to detect vertical water flow (pathways) using the analytical methods used in this manuscript. Even so, based on our data, we are convinced that there is intrapermafrost water (flow). However, we will continue to investigate this with the continuation of the monitoring data series, further boreholes at other sites and laboratory experiments, to improve our understanding of rock glacier hydrology.

---

## Author Response (AR1)

**Swiss Federal Research Institute WSL**
Eidg. Forschungsanstalt WSL
Institut fédéral de recherches WSL
Istituto federale di ricerca WSL

The Cryosphere

**Prof Adriàn Flores Orozco**

WSL Institute for Snow and Avalanche Research SLF
Alpine Environments and Natural Hazards / Permafrost
Dr Alexander Bast
Phone +41 81 4170 278
alexander.bast@slf.ch

Davos, April 29, 2024

[Figure]

[Figure]

**Response to Editor: egusphere-2024-269, submitted by Bast et al.**

Dear Professor Flores Orozco

Thank you for your positive feedback on our manuscript entitled *Short-term cooling, drying and deceleration of an ice-rich rock glacier* (egusphere-2024-269) and for the decision to accept the paper subject to minor revisions.

We have carefully considered all the comments from the two reviewers, the community and yourself. Based on the suggestions regarding the cross-borehole ERT measurements, we have decided to present the discussion in two sub-chapters: (i) discussion of water and/or ice content and its influence on rock-glacier kinematics, and (ii) discussion of ERT data quality and ERT data processing, including prospects for cross-borehole ERT measurements in permafrost environments.

Please find attached our manuscript with our suggested changes (one version with the track changes modifications visible, and one with the modifications accepted).

We hope that the revised manuscript will meet your approval and look forward to hearing from you.

With kind regards,

Alexander Bast, Marcia Phillips and Robert Kenner

---

## Editor Decision (ED1)

[revised manuscript text omitted]
 | 91 | 32 | 6 | 33 | 40 | 109 | 70 | 6 | 30 | 53 | 49 | 23 | 6 | 16 | 23 | 44 | 41 |
| HS | sd | 15 | 34 | 1 | 29 | 39 | 17 | 37 | 1 | 19 | 45 | 4 | 21 | 3 | 11 | 20 | 9 | 30 |
| HS | median | 86 | 16 | 6 | 18 | 18 | 113 | 88 | 6 | 36 | 45 | 49 | 6 | 6 | 14 | 13 | 42 | 52 |
| HS | mad | 15 | 17 | 0 | 18 | 19 | 10 | 16 | 0 | 18 | 58 | 4 | 2 | 0 | 13 | 11 | 3 | 41 |
| HS | max | 131 | 106 | 16 | 104 | 131 | 154 | 125 | 9 | 67 | 154 | 60 | 85 | 178 | 37 | 178 | 78 | 101 |
| SWE | mean | NA | NA | 2 | 92 | NA | 328 | 301 | 0 | 68 | 173 | 134 | 69 | 0 | 35 | 59 | 98 | 132 |
| SWE | sd | NA | NA | 4 | 75 | NA | 59 | 164 | 0 | 48 | 169 | 8 | 77 | 1 | 20 | 64 | 25 | 103 |
| SWE | median | NA | NA | 0 | 57 | NA | 369 | 395 | 0 | 90 | 114 | 132 | 0 | 0 | 29 | 28 | 90 | 149 |
| SWE | mad | NA | NA | 0 | 48 | NA | 15 | 33 | 0 | 40 | 169 | 11 | 0 | 0 | 16 | 41 | 10 | 128 |
| SWE | max | NA | NA | 14 | 220 | NA | 382 | 436 | 0 | 132 | 436 | 146 | 182 | 13 | 71 | 182 | 150 | 299 |

*WS - Schafberg*

[revised manuscript text omitted]

---

## Author Response (AR2)

**Swiss Federal Research Institute WSL**
Eidg. Forschungsanstalt WSL
Institut fédéral de recherches WSL
Istituto federale di ricerca WSL

The Cryosphere

**Prof Adriàn Flores Orozco**

WSL Institute for Snow and Avalanche Research SLF
Alpine Environments and Natural Hazards / Permafrost
Dr Alexander Bast
Phone +41 81 4170 278
alexander.bast@slf.ch

Davos, May 06, 2024

[Figure]

[Figure]

**Response to Editor: egusphere-2024-269, submitted by Bast et al.**

Dear Professor Flores Orozco

Thank you for your positive feedback on our manuscript entitled *Short-term cooling, drying and deceleration of an ice-rich rock glacier* (egusphere-2024-269). We are pleased that the manuscript is ready for publication, subject to some minor revisions based on your comments.

We have agreed with all your suggestions and have reworded all the sentences according to your suggestions.

Please find attached our manuscript with our changes (one version with the track changes visible, and one with the accepted changes).

We look forward to hearing from you.

With kind regards,

Alexander Bast, Marcia Phillips and Robert Kenner